# Evaluating the potential of short-term instrument deployment to improve distributed wind resource assessment

Lindsay M. Sheridan[1], Dmitry Duplyakin[2], Caleb Phillips[2], Heidi Tinnesand[2], Raj K. Rai[1], Julia E. Flaherty[1], and Larry K. Berg[1]

[1]Pacific Northwest National Laboratory, Richland, Washington, United States
[2]National Renewable Energy Laboratory, Golden, Colorado, United States

*Correspondence to*: Lindsay M. Sheridan (lindsay.sheridan@pnnl.gov)

**Abstract.** Distributed wind projects, which are connected at the distribution level of an electricity system or in off-grid applications to serve specific or local energy needs, often rely solely on wind resource models to establish wind speed and

energy generation expectations. Historically, anemometer loan programs have provided an affordable avenue for more accurate onsite wind resource assessment, and the lowering cost of lidar systems have shown similar advantages for more recent assessments. While a full twelve months of onsite wind measurement is the standard for correcting model-based long-term wind speed estimates for utility-scale wind farms, the time and capital investment involved in gathering onsite measurements must be reconciled with the energy needs and funding opportunities that drive expedient deployment of

distributed wind projects. Much literature exists to quantify the performance of correcting long-term wind speed estimates with one or more years of observational data, but few studies explore the impacts of correcting with months-long observational periods. This study aims to answer the question of how low can you go in terms of the observational time period needed to make impactful improvements to model-based long-term wind speed estimates.

Three algorithms, multivariable linear regression, adaptive regression splines, and regression trees, are evaluated for their

skill at correcting long-term wind resource estimates from the European Centre for Medium-Range Weather Forecasts Reanalysis version 5 (ERA5) using months-long periods of observational data from 66 locations across the United States. On average, correction with even one month of observations provides significant improvement over the baseline ERA5 wind speed estimates and produces median bias magnitudes and relative errors within 0.22 m s$^{-1}$ and four percentage points of the median bias magnitudes and relative errors achieved using the standard twelve months of data for correction. However, in

cases when the shortest observational periods (one to two months) used for correction are not well correlated with the overlapping ERA5 reference, the resultant long-term wind speed errors are worse than those produced using ERA5 without correction. Summer months, which are characterized by weaker relative wind speeds and standard deviations for most of the evaluation sites, tend to produce the worst results for long-term correction using months-long observations. The three tested algorithms perform similarly for long-term wind speed bias; however, regression trees perform notably worse than

multivariable linear regression and adaptive regression splines in terms of correlation when using six months or less of observational data for correction.

Translating the analysis to wind energy, median relative errors in the capacity factor are on average within 10% using one month of training. If the observation period used for correction is not well correlated with the reference data, however, misrepresentation of the observed capacity factor can be substantial. The risk associated with poor correlation between the observed and reference datasets decreases with increasing training period length. In the worst correlation scenarios, the median capacity factor relative errors from using one, three, and six months are within 47%, 26%, and 16%, respectively.

## 1 Introduction

In the utility-scale wind energy industry, short-term (less than five years) wind measurements are temporally extended using long-term (decades-long) wind resource simulations to produce a long-term wind energy generation estimate at a site of development interest in an expedient manner. Wind farm project planners install onsite wind measurement instruments (e.g., meteorological towers, lidars, sodars) to gather data, but cannot wait decades to establish long-term wind resource characterization based on such measurements alone (Lackner et al., 2008). Instead, wind analysts commonly simulate the long-term wind resource by correlating the short-term onsite wind measurements with long-term reference data, such as atmospheric reanalyses, high-resolution mesoscale models, or other nearby measurements using a measure-correlate-predict (MCP) approach.

Distributed wind projects, particularly those involving small wind turbines, are more subject to challenging financial, spatial, and time constraints than utility-scale wind. Onsite wind measurements are often not feasible or economically viable investments, leading to many distributed wind projects that relied solely on wind resource models to establish generation estimates, which are helpful but not entirely accurate. Between 2000 and 2011, the U.S. Department of Energy (DOE)'s Wind Powering America initiative sponsored an Anemometer Loan Program to provide a more affordable avenue for onsite wind resource assessment, which resulted in the installation of 128 anemometers across the U.S. and a template for around 20 state administered anemometer loan programs (Jimenez, 2013). More recently, researchers from Aurora College have begun deploying mobile lidar systems in northern Canada to study the viability of wind energy in communities that are reliant on diesel, which is expensive and difficult to transport to remote locations (Seto, 2022). In the U.S., communities that receive DOE technical assistance to transform their energy systems, such as through the Energy Transitions Initiative Partnership Project (DOE, 2023), continue to weigh the costs and benefits of onsite wind measurement. In addition to concerns regarding capital investment, communities must often reconcile the time investment involved for gathering onsite measurements with the energy needs and funding opportunities driving expedient deployment of wind turbines.

The vast majority of wind resource assessment literature supports collecting at least one year of onsite measurements to represent a full seasonal wind cycle, including the analyses of Dinler (2013), Liléo et al. (2013), Mifsud et al. (2018), Zakaria et al. (2018), Tang et al. (2019), and Chen et al. (2022). Miguel et al. (2019) found that uncertainty in long-term wind resource estimates reduced by 18%, 29%, 35%, and 40% when one, two, three, and four years of wind measurements, respectively, were added to the monitoring campaign. Additionally, private companies that specialize in providing resource

assessment for wind projects also tend to require at least one year of onsite measurements to characterize the wind. For example, ArcVera uses at least one full year of observational data to bias correct their high-resolution model output (ArcVera, 2023). In their discussion of wind resource assessment, wind measurement company NRG Systems (2023) states that measurements of meteorological parameters are typically taken over the course of several years at a potential wind farm site using a combination of meteorological towers and lidars.

A small number of studies, however, explore adjusting long-term predictions using observational data with less than one year of temporal coverage. Taylor et al. (2004 as reported via Carta et al., 2013), Weekes and Tomlin (2014), and Basse et al. (2021) reported that wind speed errors were within 4%, 4.8%, and 4%, respectively, using three months of onsite measurements. Using data from the United Kingdom, Derrick (1992 as reported via Rogers et al., 2005) found that eight months of onsite data were needed to minimize uncertainties. The performance of long-term predictions using less than one year of on-site measurements varied according to the season(s) during which the onsite measurements were taken. For their experiment in the United Kingdom, Weekes and Tomlin (2014) found that the smallest errors occurred when using measurements taken in early spring or fall. Basse et al. (2021) performed MCP tests in Germany using linear regression and variance ratio algorithms and found that using variance ratio produced overestimates when using summer measurements and underestimations when using winter measurements, with the opposite trend noted for linear regression.

The present study expands previous analyses of long-term wind resource performance based on months-long observations to diverse locations across the United States, with a focus on measurement heights relevant to distributed wind installations (20 m – 100 m). We explore multiple MCP algorithm options and highlight the best performers for generating accurate long-term wind speed estimates for a variety of error metrics relevant to the wind energy industry. Section 2 describes the wind measurements employed as 1) the months-long training input for MCP-based long-term wind estimates and 2) the long-term datasets to validate the MCP results. The reference dataset, ERA5, is also discussed in Section 2, along with three MCP algorithms selected for evaluation and the error metrics employed to test their performance. Section 3 provides the performance analysis of MCP-based long-term wind resource estimation with less than one year of onsite measurements with a focus on establishing the minimum number of months needed for certain levels of accuracy. Additionally, Section 3 relates the performance results to a variety of influences, including geographical location and the time of year that measurements were gathered. Section 4 summarizes the performance analysis and relates the wind speed error metrics to impacts on wind energy generation expectations.

## 2 Data discussion and methodology

The MCP model for long-term wind resource assessment begins with establishing a short-term relationship between observational wind data measured at a target site and concurrent data at a nearby reference site. The reference data can be observations or model data and typically includes wind variables (speed and direction) and potentially additional relevant meteorological (temperature, pressure) or temporal (hour of day, month of year) variables. The resultant short-term

relationship is subsequently applied to long-term data from the reference site to predict the long-term wind resource at the target site (Rogers et al., 2005).

## 2.1 Wind observations

The 66 wind speed observational datasets in this analysis are sourced from U.S. Department of Energy national laboratories, facilities, and projects, the National Data Buoy Center, the Bonneville Power Administration, and the Forest Ecosystem Monitoring Cooperative. One observational dataset was collected using a lidar and the remaining measurements were gathered from anemometers on meteorological towers. Most of the observational collection is publicly available (57 sites), while a small number of datasets are subject to non-disclosure agreements (9 sites) as outlined in the data availability statement.

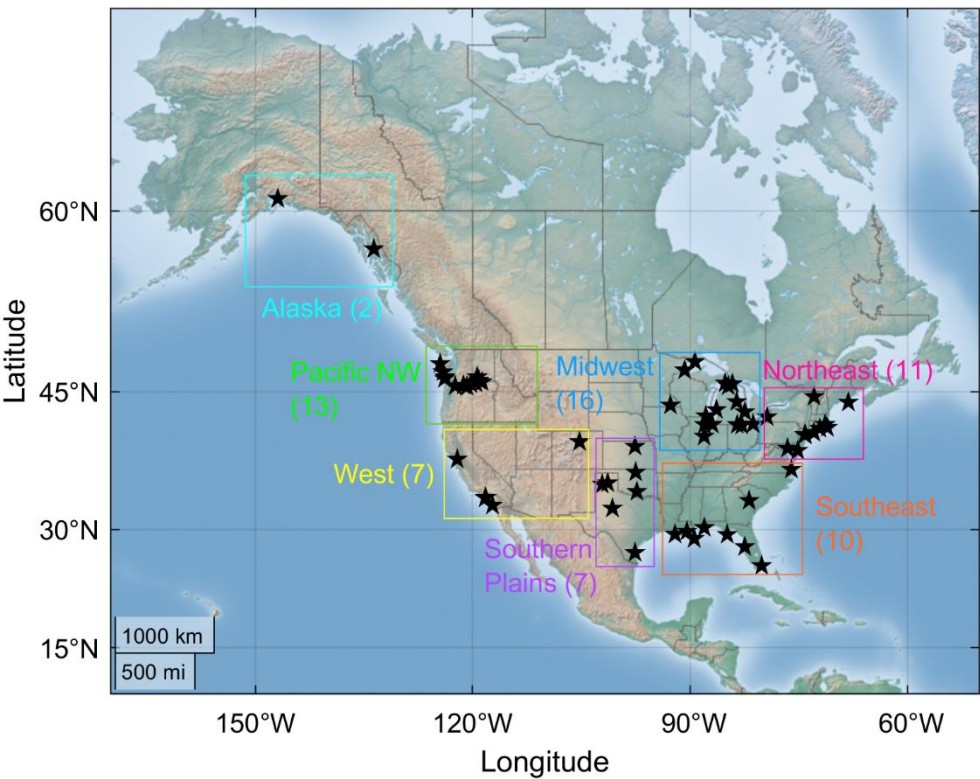

**Figure 1.** Locations of wind measurements assessed for establishing long-term performance based on months-long observations used in this study. The number of observational sites per region is included in parentheses.

The measurement sites utilised in this work span 28 states and are diverse in terrain complexity and land cover (Figure 1). The measurement heights are similarly diverse, from 20 m to 100 m (Figure 2a), representing a wide range of the distributed wind hub heights that are reported in the United States (PNNL, 2024). Many of the lowest observations, which align with small distributed wind turbine hub heights (between 20 m and 40 m), source from the National Data Buoy Center and are

located along coastlines. The highest observations, which align with large distributed wind turbine hub heights (between 80 m and 100 m), are in Long Island, New York (85 m) and the San Francisco Bay Area, California (100 m).

To establish reliable datasets for MCP training and validation, the wind speed observations were quality-controlled by removing instances or periods of atypical or unphysical reported wind speeds (less than 0 m s⁻¹, greater than 50 m s⁻¹, or nonvarying periods of time greater than four hours) that might indicate instrument error due to an outage or weather impacts such as icing. The temporal coverage of the wind speed observations at the measurement sites ranged from 3.5 – 23 years, with an average of 11 years (Figure 2b). The long-term measurement wind speeds ranged from 2.0 m s⁻¹ to 9.1 m s⁻¹ with an average across all measurement sites of 5.6 m s⁻¹.

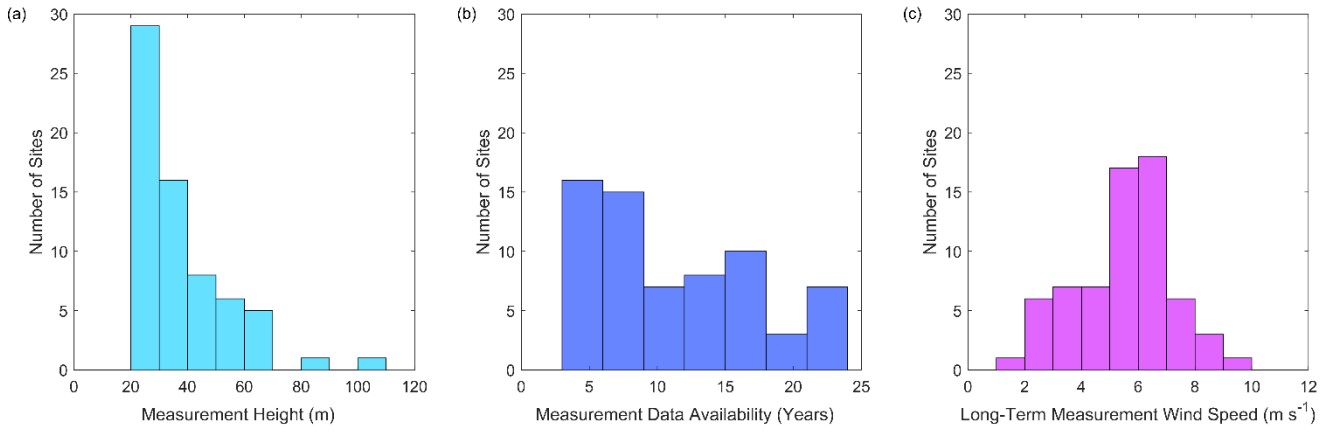

**Figure 2.** (a) Measurement heights, (b) long-term measurement availability, and (c) long-term measurement wind speeds for the sites evaluated for long-term performance based on months-long observations.

## 2.2 Reanalysis model for long-term correction

ERA5 is a popular global reanalysis model (Hersbach et al., 2020) utilised for wind energy resource assessments in a variety of ways, including as a standalone product, as input boundary conditions to higher-resolution model runs, and as a reference dataset for MCP with local observations (Olauson, 2018; Soares et al., 2020; Hayes et al., 2021; de Assis Tavares et al., 2022). ERA5 is developed by the European Centre for Medium-Range Weather Forecasts and provides meteorological data from 1950-present at 1-hour temporal resolution with a horizontal grid spacing of 0.25° (Hersbach et al., 2020). At many locations where validation has been performed, ERA5 tends to produce high Pearson correlation coefficients (Eq. 1) and negative biases (indicating underestimation) (Eq. 2) between the simulated ($u_{sim}$) and observed ($u_{obs}$) wind speeds. Ramon et al. (2019) utilised 77 meteorological towers around the globe with measurement heights ranging from 10 m to 122 m and found median ERA5 seasonal wind speed biases between 0 and -1 m s⁻¹, though ERA5 had the best correlation with observations among five reanalyses. Across 62 sites in the continental US, Sheridan et al. (2022) found that ERA5 underestimated the observed wind speeds by an average of 0.5 m s⁻¹ but had higher correlations (average of 0.77) than two alternate reanalyses and wind models. Using measurements from more than 100 onshore and offshore lidars, sodars, and meteorological towers across the United States, Wilczak et al. (2024) determined that ERA5-derived wind power estimates

were biased low by 20%. At locations across Europe, Murcia et al. (2022) determined that ERA5 slightly underestimated the observed wind speeds (average bias = -0.06 m s$^{-1}$) and provided a high degree of correlation (average of 0.92).

### 2.3 Metrics for performance evaluation

This study aims to reduce the ERA5-based wind speed bias using months-long onsite observations while not degrading other metrics of error, such as correlation. The Pearson correlation coefficient explains the degree to which the simulated and observed wind speeds are linearly related, with values close to one indicating a high degree of correlation (Eq. 1). The wind speed bias, i.e., the average difference between the simulated ($U_{sim}$) and observed ($U_{obs}$) wind speeds over a time series of length $N$, informs on whether a simulation tends to overestimate (positive bias), underestimate (negative bias), or accurately

represent (zero bias) the observed wind resource (Eq. 2). This work also considers the bias magnitude (the absolute value of bias (Eq. 2)) when comparing multiple sites, as combinations of positive and negative biases can obscure the degree of error. The relative error in the long-term wind speed simulation is the absolute difference between the simulated and observed wind speeds normalized by the observed wind speed, providing insight on the magnitude of error in a simulation (Eq. 3).

$$correlation = \frac{\sum_{i=1}^{N}(U_{sim,i} - \overline{U_{sim}})(U_{obs,i} - \overline{U_{obs}})}{\sqrt{\sum_{i=1}^{N}(U_{sim,i} - \overline{U_{sim}})^2}\sqrt{\sum_{i=1}^{N}(U_{obs,i} - \overline{U_{obs}})^2}} \tag{1}$$

$$bias = \frac{1}{N}\sum_{i=1}^{N}(U_{sim,i} - U_{obs,i}) \tag{2}$$

$$relative\ error = 100\% * \frac{|\overline{U_{sim}} - \overline{U_{obs}}|}{\overline{U_{obs}}} \tag{3}$$

To set a baseline of ERA5 performance for the suite of observations utilised in this work, we adjust the ERA5 wind speeds to each measurement height $z$ (Figure 2a) using the power law (Eq. 4) with the shear exponent α calculated at each timestamp using the ERA5 wind speeds at 10 m and 100 m (Eq. 5). Horizontally, we adjust the ERA5 wind speeds to the observational location using inverse distance-weighted interpolation.

$$U_{era5\_z} = U_{era5\_100m}\left(\frac{z}{100\ m}\right)^{\alpha} \tag{4}$$

$$\alpha = \frac{\ln(U_{era5\_100m}/U_{era5\_10m})}{\ln(100\ m/10\ m)} \tag{5}$$

In all regions except for the Southeast, the ERA5 wind speeds tend to be lower than the observed wind speeds (Figure 3a). The underestimation is most pronounced in Alaska (median bias = -1.61 m s$^{-1}$), the Pacific Northwest (-1.18 m s$^{-1}$), and

the Southern Plains (-1.74 m s$^{-1}$). The same three regions have the highest relative errors (medians = 28%, 25%, and 20%, respectively) (Figure 3b). The median correlations exceed 0.75 for all regions except Alaska (0.69) and the West (0.67), where the coarse ERA5 is challenged by mountainous and coastal terrain (Figure 3c). No consistent trends in ERA5 performance are noted according to height above ground (Figure 3d, e, f). The wind speed relative errors are greatest for measurement heights between 30 m and 40 m (median = 31%), while the median relative errors for measurement heights between 1) 20 m and 30 m and 2) 40 m and 50 m are 11% and 10%, respectively. Across all sites, the median statistics are: bias = -0.50 m s$^{-1}$, bias magnitude = 0.67 m s$^{-1}$, relative error = 13%, and correlation = 0.78. The tendencies of ERA5 to underestimate the observed wind speeds in this analysis while exhibiting a relatively high degree of correlation with them aligns with the findings of Ramon et al. (2019), Murcia et al. (2022), Sheridan et al. (2022), and Wilczak et al. (2024) discussed in Section 2.2. The bias trends according to region (Figure 3a) also align with the findings of Wilczak et al. (2024) in that ERA5 underestimation is noted in the Pacific Northwest and Southern Plains, while a mix of overestimation and underestimated is noted for the Midwest.

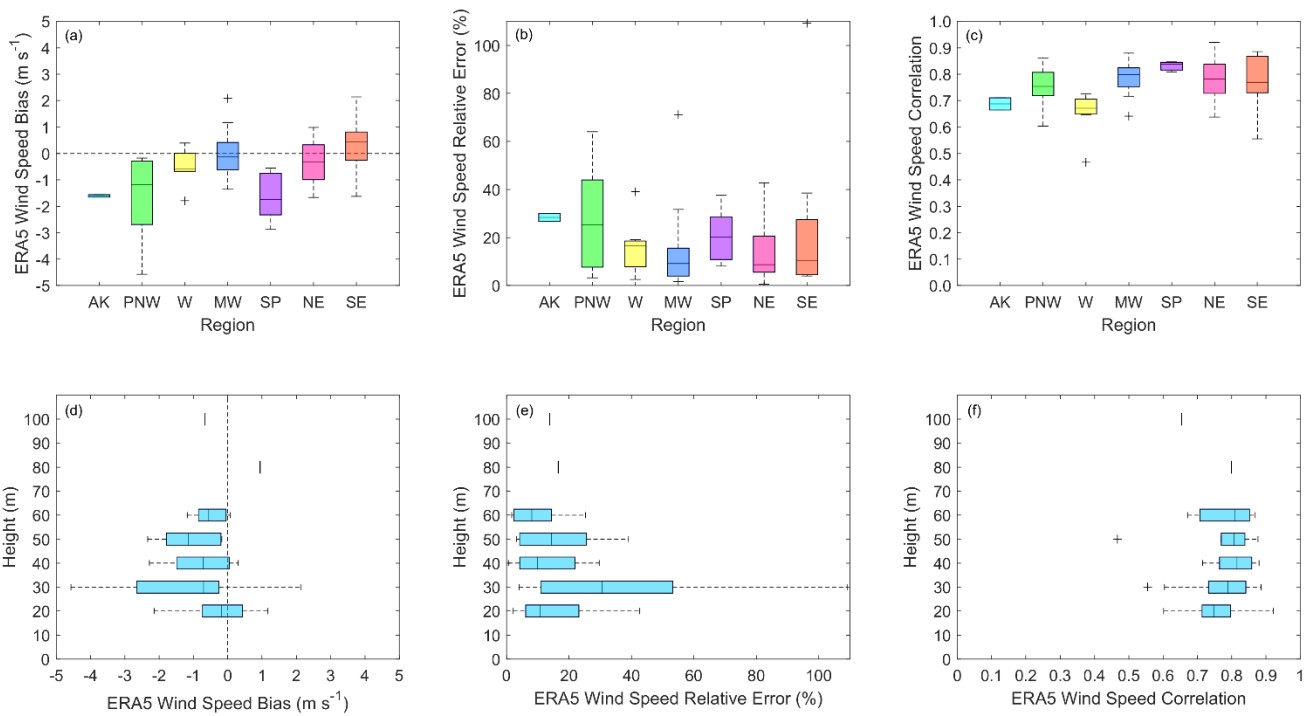

**Figure 3.** Long-term ERA5 wind speed (a), (d) bias (b), (e) relative error, and (c), (f) correlation across 66 measurement sites in the United States, grouped by region (top) and measurement height (bottom). AK = Alaska, PNW = Pacific Northwest, W = West, MW = Midwest, SP = Southern Plains, NE = Northeast, and SE = Southeast.

**2.4 MCP methodologies**

One of the advantages of utilising MCP for long-term wind resource estimation is the variety of algorithm choices, which range from simplistic linear regression to machine learning techniques that can be applied to link the short-term and long-

term wind speeds. Early MCP methodologies focused on linear (as reported via Rogers et al., 2005: Derrick, 1992; Landberg and Mortenson, 1993; Woods and Watson, 1997; Vermeulen et al., 2001) and quadratic fits (as reported via Rogers et al., 2005: Joensen et al., 1999; Riedel and Strack, 2001). From there, distribution-based probabilistic techniques emerged (García-Rojo, 2004; Sheppard, 2009; Carta and Velázquez, 2011). With the onset of machine learning techniques came applications to MCP-based wind resource analysis, such as using artificial neural networks, support vector machine, and random forest to estimate long-term wind speeds (Díaz et al., 2017).

This analysis evaluates three algorithms for their skill at creating long-term wind resource estimates based on varying temporal lengths of observational data: (1) multiple linear regression (MLR), (2) adaptive regression splines (ARS), and (3) regression trees (RT). These algorithms were selected because they are broadly available and represent diversity in complexity and approach. Multiple linear regression estimates the linear relationship between a target variable (onsite wind measurements in this analysis) and more than one reference variable (sourced from ERA5 in this analysis). Adaptive regression splines involve the construction of piecewise-cubic regression models based on the short-term target and reference datasets (Jekabsons, 2016). In this analysis, we utilise the default parameter configurations of Jekabsons (2016). The maximum number of basis functions follows the formula of Milborrow (2016): min(200, max(20, 2*the number of input variables)) + 1. The maximum degree of interactions between input variables is set to 1 for additive modelling, therefore the generalized cross-validation penalty per knot is set to 2 following the recommendation of Friedman (1991). Regression trees recursively partition and evaluate the concurrent short-term target and reference datasets into unique segments, which are subsequently used to predict long-term target behaviour. In this analysis, the ensemble aggregation method used is least-squares boosting with 100 learning cycles per the Matlab algorithm fitrensemble (MathWorks, 2024). Numerous additional algorithms have been developed and tested for their ability to improve simulation accuracy, and it is important to note that each feature different approaches, computational investments, complexities, skills, and limitations. For example, Rogers et al. (2005) note that linear regression techniques are easily implemented and well suited for performing bias correction but have a tendency to create a bias in the variance that variance-conserving MCP techniques are better suited to resolve.

To narrow down an effective training approach, different combinations of reference variables are explored for their impact on long-term wind resource assessment error metrics. The analysis of Phillips et al. (2022) identified reanalysis wind speed, reanalysis wind direction, and time of day as the most important variables for wind speed bias correction using a variety of techniques including multivariable linear regressions and regression trees. Therefore, we explore progressively increasing variable combinations of ERA5 wind speeds at the provided output heights of 10 m and 100 m ($U_{era5\_10m}$ and $U_{era5\_100m}$), power law-based wind speed estimates at the measurement height $z$ ($U_{era5\_z}$) (Eq. 4) using the shear exponent $\alpha$ based on the ERA5 wind speeds at 10 m and 100 m (Eq. 5), ERA5 $u$ and $v$ wind components at 10 m and 100 m ($u_{era5\_10m}$, $v_{era5\_10m}$, $u_{era5\_100m}$, and $v_{era5\_100m}$), and the hour of the day.

As an initial test of the performance of the algorithms and reference variable combinations, we develop ensembles of MCP-based long-term wind speed estimates at each measurement site using consecutive 12-month training periods, according to the following steps:

1. Establish that 75% of the observations in each month in the training period are available after applying the quality control checks discussed in Section 2.1 (all 66 observations utilised in this work have average and median monthly data recovery and quality rates exceeding the 75% threshold).
2. A model is trained on temporally aligned observation data and reference data during the training period.
3. The model is used to predict the full observation period (Figure 2b).
4. Performance statistics are computed with respect to the observations (Table 1). Timestamps with missing observations are excluded from the statistics.

**Table 1.** Long-term MCP-based wind speed error metrics averaged (median) across 66 sites using different combinations of algorithms and training variables. Values in bold indicate the optimal combination of training variables for each error metric and algorithm.

| Error Metric | Algorithm | Training Variables | | | |
|---|---|---|---|---|---|
| | | $U_{era5\_10m}$, $U_{era5\_100m}$ | $U_{era5\_10m}$, $U_{era5\_100m}$ $U_{era5\_z}$ | $U_{era5\_10m}$, $U_{era5\_100m}$ $U_{era5\_z}$ $u_{era5\_10m}$, $v_{era5\_10m}$ $u_{era5\_100m}$, $v_{era5\_100m}$ | $U_{era5\_10m}$, $U_{era5\_100m}$ $U_{era5\_z}$ $u_{era5\_10m}$, $v_{era5\_10m}$ $u_{era5\_100m}$, $v_{era5\_100m}$ *hour of day* |
| Bias Magnitude (m s⁻¹) | MLR | 0.09 (0.08) | 0.09 (0.08) | **0.08 (0.07)** | **0.08 (0.07)** |
| | ARS | 0.09 (0.08) | 0.09 (0.08) | **0.08 (0.07)** | **0.08 (0.07)** |
| | RT | 0.09 (0.08) | 0.09 (0.08) | 0.08 (0.07) | **0.08 (0.06)** |
| Correlation | MLR | 0.78 (0.80) | 0.78 (0.80) | 0.80 (0.81) | **0.81 (0.82)** |
| | ARS | 0.78 (0.80) | 0.78 (0.80) | 0.82 (0.83) | **0.82 (0.84)** |
| | RT | 0.74 (0.76) | 0.74 (0.76) | 0.78 (0.80) | **0.79 (0.80)** |

All MCP combinations of algorithms and training variables provide substantially improved long-term bias magnitudes compared to the ERA5 (i.e., $U_{era5\_z}$) average (median) bias magnitude of 1.01 m s⁻¹ (0.67 m s⁻¹) (Table 1). With 12 months of training time, the MCP average and median bias magnitudes vary by at most 0.02 m s⁻¹ according to algorithm and training variables. More variability is seen according to the various combinations of MCP algorithms and training variables for correlation (Table 1). Using just $U_{era5\_10m}$ and $U_{era5\_100m}$ as training variables, MLR and ARS improve on the ERA5 average (median) correlation of 0.76 (0.77), while RT produces a lower correlation. Utilising all training variables generates the highest overall correlations (Table 1). Given the optimal correlation results found when training for 12 months with MCP using the complete variable set of ERA5 wind speeds, ERA5 *u* and *v* components, and hour of the day, these variables are selected for evaluating long-term MCP wind speed estimates using months-long training periods. Using 12 months of observations, the three algorithms perform similarly for bias and ARS is the best performing algorithm for correlation. The following sections explore whether that status holds when using months-long training periods.

The months-long analysis follows the same ensemble formula as the 12-month exercise, just with shorter consecutive training periods. Across the measurement sites, calendar months in the spring and fall had the most single instances of ≥75% data recovery and quality, followed by summer, and lastly winter (Figure 4). Median measurement data recovery and quality percentages according to calendar month ranged from 99.2% (December) to 99.7% (May) (Figure 4).

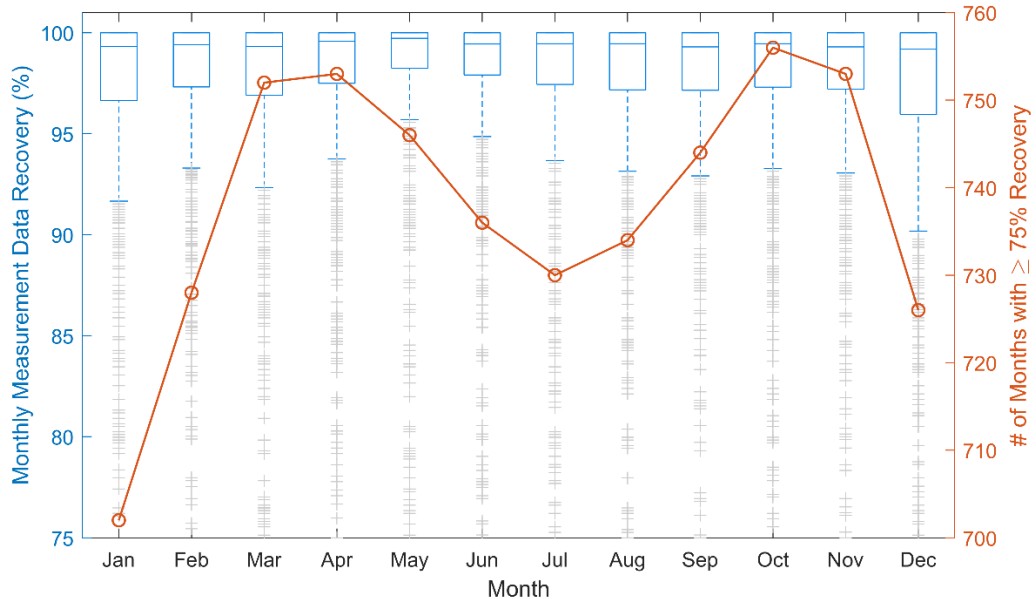

**Figure 4.** Monthly measurement data recovery and quality (boxplot) and number of months with ≥ 75% data recovery.

## 3 Results and Discussion

### 3.1 Long-term wind estimation performance according to length of training period

On average, even one month of observations combined with any of the three assessed MCP algorithms provides substantial improvement over using ERA5 wind speeds alone for long-term wind speed estimation. Figure 5a displays the ensemble average bias magnitude at each site using increasing numbers of training months. The median bias magnitudes across all sites using one month of training are 0.28 m s$^{-1}$ (MLR) and 0.29 m s$^{-1}$ (ARS, RT), as compared with 0.67 m s$^{-1}$ using ERA5 (Table 2, Figure 5a). The median bias magnitudes drop to 0.18 m s$^{-1}$ using three months of training and 0.12 m s$^{-1}$ using six months of training. For all training durations, the MCP algorithms perform within 0.01 m s$^{-1}$ of each other for median bias magnitude. Similarly, the relative errors in the long-term wind speed estimates decrease substantially from those using ERA5 (median = 13%) (Figure 5b). One month of training produces median relative errors of 6% across the algorithms, which decreases to 4% with three months of training and 2-3% with six months of training (Table 2). The standard deviations of bias magnitude provide an indication of the uncertainty of the MCP-based long-term wind speed estimates, ranging from

0.24 m s$^{-1}$ (MLR, RT) to 0.28 m s$^{-1}$ (ARS) using one month of training, 0.14 m s$^{-1}$ (RT) to 0.15 m s$^{-1}$ (MLR, ARS) using three months of training, and 0.08 m s$^{-1}$ (ARS, RT) to 0.09 m s$^{-1}$ (MLR) using six months of training (Figure 5c, Table 2).

**Table 2.** Median biases, bias magnitudes, standard deviations of bias magnitudes, relative errors, and correlations according to algorithm and number of training months.

| | | Number of Training Months | | | | | | | | | | | |
|---|---|---|---|---|---|---|---|---|---|---|---|---|---|
| **Error Metric** | **Algorithm** | **1** | **2** | **3** | **4** | **5** | **6** | **7** | **8** | **9** | **10** | **11** | **12** |
| **Bias (m s$^{-1}$)** | MLR | -0.03 | -0.02 | -0.02 | -0.01 | 0.00 | 0.00 | 0.00 | 0.01 | 0.01 | 0.01 | 0.01 | 0.01 |
| | ARS | -0.04 | -0.02 | -0.02 | -0.01 | 0.00 | 0.00 | 0.01 | 0.01 | 0.01 | 0.01 | 0.01 | 0.01 |
| | RT | -0.08 | -0.04 | -0.03 | -0.01 | -0.01 | 0.00 | 0.00 | 0.01 | 0.00 | 0.00 | 0.00 | 0.00 |
| **Bias Magnitude (m s$^{-1}$)** | MLR | 0.28 | 0.21 | 0.18 | 0.16 | 0.14 | 0.12 | 0.11 | 0.10 | 0.09 | 0.08 | 0.07 | 0.07 |
| | ARS | 0.29 | 0.22 | 0.18 | 0.15 | 0.13 | 0.12 | 0.10 | 0.09 | 0.08 | 0.08 | 0.07 | 0.07 |
| | RT | 0.29 | 0.22 | 0.18 | 0.16 | 0.13 | 0.12 | 0.10 | 0.09 | 0.08 | 0.07 | 0.07 | 0.07 |
| **Standard Dev. of Bias Magnitude (m s$^{-1}$)** | MLR | 0.24 | 0.18 | 0.15 | 0.13 | 0.11 | 0.09 | 0.08 | 0.07 | 0.06 | 0.06 | 0.05 | 0.05 |
| | ARS | 0.28 | 0.19 | 0.15 | 0.12 | 0.10 | 0.08 | 0.07 | 0.07 | 0.06 | 0.05 | 0.05 | 0.05 |
| | RT | 0.24 | 0.18 | 0.14 | 0.12 | 0.10 | 0.08 | 0.07 | 0.06 | 0.06 | 0.05 | 0.05 | 0.05 |
| **Relative Error (%)** | MLR | 5.7 | 4.5 | 3.8 | 3.2 | 2.8 | 2.6 | 2.4 | 2.1 | 1.8 | 1.6 | 1.4 | 1.3 |
| | ARS | 5.7 | 4.4 | 3.6 | 3.1 | 2.9 | 2.6 | 2.2 | 1.9 | 1.7 | 1.5 | 1.3 | 1.3 |
| | RT | 5.6 | 4.3 | 3.7 | 3.1 | 2.8 | 2.4 | 2.1 | 1.8 | 1.6 | 1.4 | 1.3 | 1.2 |
| **Correlation** | MLR | 0.79 | 0.80 | 0.81 | 0.81 | 0.81 | 0.81 | 0.81 | 0.82 | 0.82 | 0.82 | 0.82 | 0.82 |
| | ARS | 0.77 | 0.80 | 0.81 | 0.82 | 0.83 | 0.83 | 0.83 | 0.83 | 0.83 | 0.84 | 0.84 | 0.84 |
| | RT | 0.67 | 0.70 | 0.72 | 0.74 | 0.76 | 0.77 | 0.78 | 0.78 | 0.79 | 0.80 | 0.80 | 0.80 |

Considering the sign of the bias, we recall that ERA5 tends to underestimate the observed wind speeds with a median bias of -0.50 m s$^{-1}$ across the sites. On average, incorporating months-long observations into the long-term estimations moves the bias substantially closer to zero. Applying just one month of observations results in median biases between -0.08 m s$^{-1}$ (RT) and -0.03 m s$^{-1}$ (MLR). For all training period lengths of four months or greater, the median MCP-based biases are within ±0.01 m s$^{-1}$, regardless of algorithm (Table 2).

The MCP algorithms diverge in their performance for long-term wind speed correlation, especially when using four or fewer training months. Using RT with a limited number of training months does not improve correlation relative to ERA5 and the other MCP algorithms. Using one month of training, MLR and ARS produce similar correlations (medians = 0.79 and 0.77, respectively) to ERA5 (median = 0.78), while the RT correlations are quite a bit worse (median = 0.67) (Table 2, Figure 5d). Only when the training period is at least seven months do the median RT correlations match ERA5. An interesting correlation comparison is noted for the National Renewable Energy Laboratory's National Wind Technology Center, located in an extremely windy corridor of the complex terrain along Colorado's Front Range. All of the lowest correlation outliers in Figure 5d are from this site and, while MLR and ARS match or improve upon the ERA5 correlation for the National Wind Technology Center (0.47) using any training period length of at least two months, RT never achieves

the ERA5 correlation even with a full twelve months of training. For all training durations of at least four months, ARS is on average the best performing MCP algorithm for correlation.

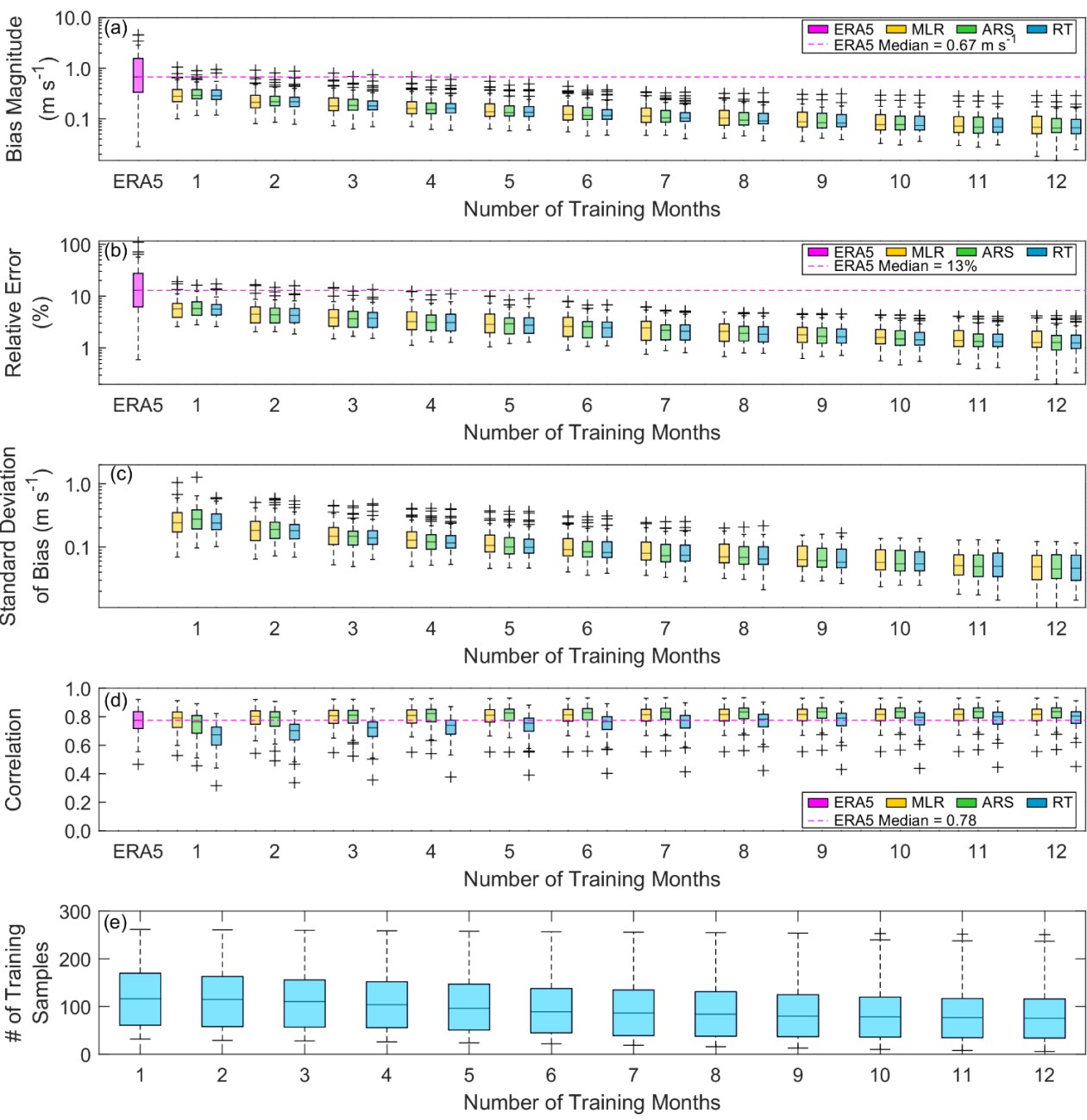

**Figure 5.** Average long-term (a) bias magnitude, (b) relative error, (c) standard deviation of bias magnitude, and (d) correlation for 66 sites comparing observations with ERA5 and MCP techniques using varying training period lengths, along with (e) the number of training samples per site and per number of training months.

Given that a twelve-month training period is most commonly employed for long-term wind speed estimation using MCP, it is beneficial to consider MCP performance using less than one year of observations relative to the twelve-month standard. Using a one-month training period, the median long-term bias magnitudes (relative errors) for 66 sites are within 0.22 m s$^{-1}$ (4 percentage points) of the median long-term bias magnitudes (relative errors) determined using twelve months regardless of MCP algorithm (Figure 6a,b). However, at one coastal Alaskan site the difference in bias magnitude (relative error) when using one versus twelve months of observations can reach 1 m s$^{-1}$ (18%). Training with three months of observations places the median bias magnitudes (relative errors) within 0.12 m s$^{-1}$ (3 percentage points) of the median bias magnitudes (relative errors) established when using twelve months, with outlier differences within 0.7 m s$^{-1}$ (13%). Using six-month training periods results in median bias magnitudes (relative errors) within 0.05 m s$^{-1}$ (1 percentage point) of the twelve-month median bias magnitudes (relative errors), with outlier differences within 0.4 m s$^{-1}$ (6%).

Great disparity is noted among the MCP algorithms when comparing their correlation performance using months-long training periods versus training periods of one-year duration (Figure 6c). Using one month of training versus twelve months produces median correlation differences of 0.03 for MLR, 0.07 for ARS, and 0.13 for RT. With six-month training periods, MLR and ARS produce a median correlation less than 0.01 different than using twelve months, while the six-month median RT correlation is 0.04 different from using twelve months.

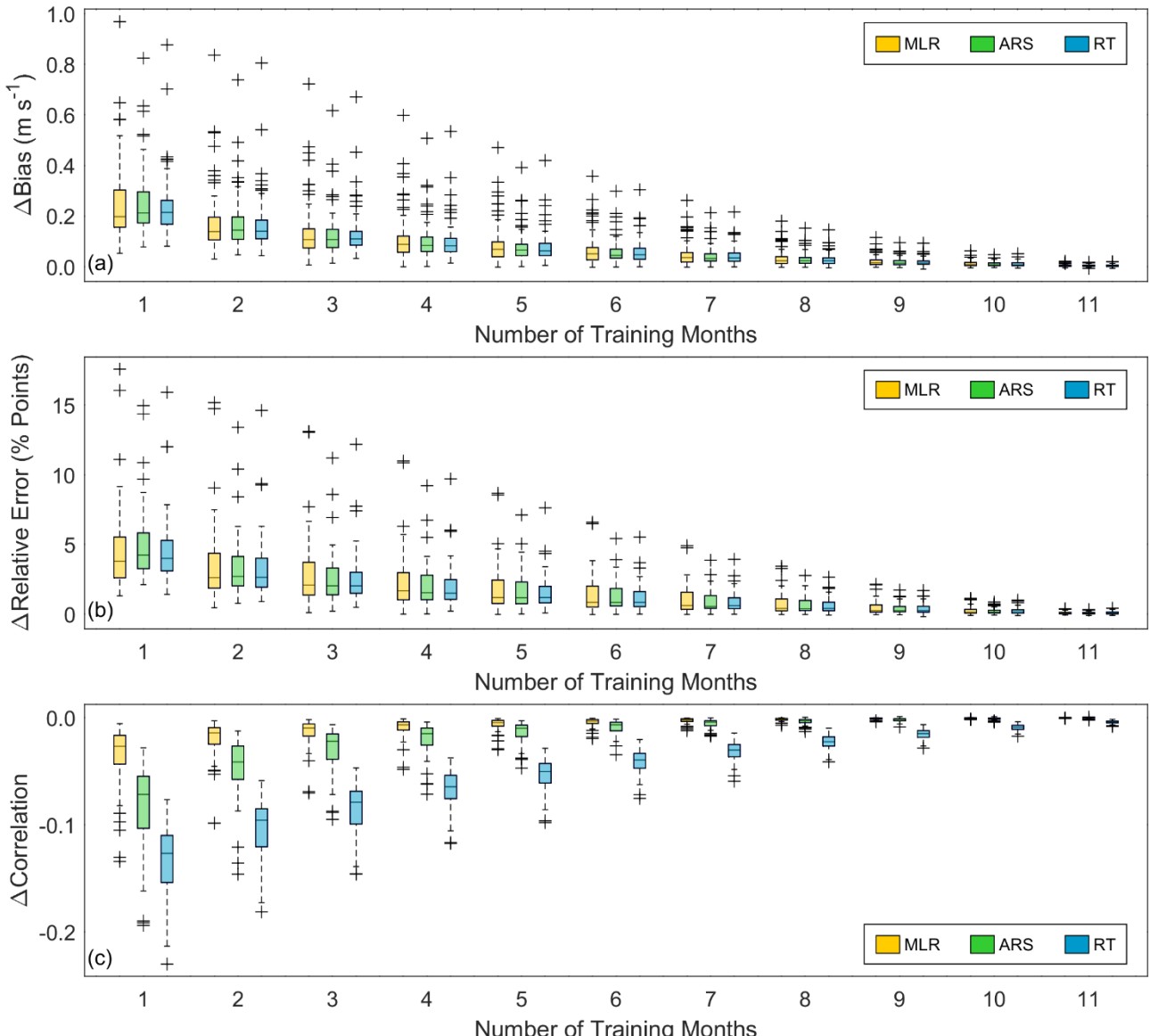

**Figure 6.** Difference in average long-term (a) bias magnitude, (b) relative error, and (c) correlation for 66 sites between MCP training periods of one to eleven months and MCP training periods of twelve months.

The results in Figure 5 and Figure 6 explain the degree of error expected on average when using MCP with observations less than one year in duration, highlighting similar performance across the MCP algorithms for long-term wind speed bias magnitude, but less successful performance when using RT with months-long observations for correlation. However, since wind measurement timelines tend to be more dependent on funding availability, project deadlines, and fair-weather deployment windows than on identifying the most representative period for long-term wind representation, it is imperative to

consider the worst-case scenario errors (Figure 7). In this scenario, we identify the worst-performing ensemble member for each error metric (largest bias magnitude, largest MAE, smallest correlation) and each algorithm (MLR, ARS, RT) according to length of training period for each of the 66 sites. It is important to keep in mind that the worst-case scenario error analysis

is a conservative approach that is not analogous to assessing algorithm uncertainty. Additionally, more robust algorithms than those studied in this work could reduce the sensitivity to the outliers in the shortest training timeseries that drive error in the long-term estimates.

In the worst-case scenario, using a single month for MCP training will produce long-term errors significantly worse than simply using ERA5 to produce long-term wind speed estimates (Figure 7). Despite its performance success on average

(Figure 5), ARS produces the largest bias magnitudes (median = 1.36 m s$^{-1}$), the largest relative errors (median = 27%), and the smallest correlations (median = 0.20) in the worst-case scenario. MLR and RT perform similarly for bias using a one-month training period in the worst-case scenario (median bias magnitudes = 1.23 m s$^{-1}$ and 1.18 m s$^{-1}$, respectively; median relative errors = 22% for both algorithms), while MLR performs best in terms of correlation (median = 0.63).

A training period of four months provides improvement in bias magnitude and relative error over simply using ERA5

(medians = 0.67 m s$^{-1}$ and 13%, respectively) for long-term wind resource estimation, even in the worst-case scenario, regardless of MCP algorithm choice (medians = 0.53 m s$^{-1}$ – 0.59 m s$^{-1}$ and 13%, respectively) (Figure 7). For correlation, MLR exceeds ERA5 (median = 0.78) at five months, ARS at six months, and RT at twelve months (medians = 0.79). Though all three algorithms provide similar improvement in relative errors and MLR and ARS provide the most beneficial correlations on average when using MCP with months-long observations (Figure 5, Table 2), MLR is the least risky

approach given the possibility of measurement during unfavourable wind conditions for MCP.

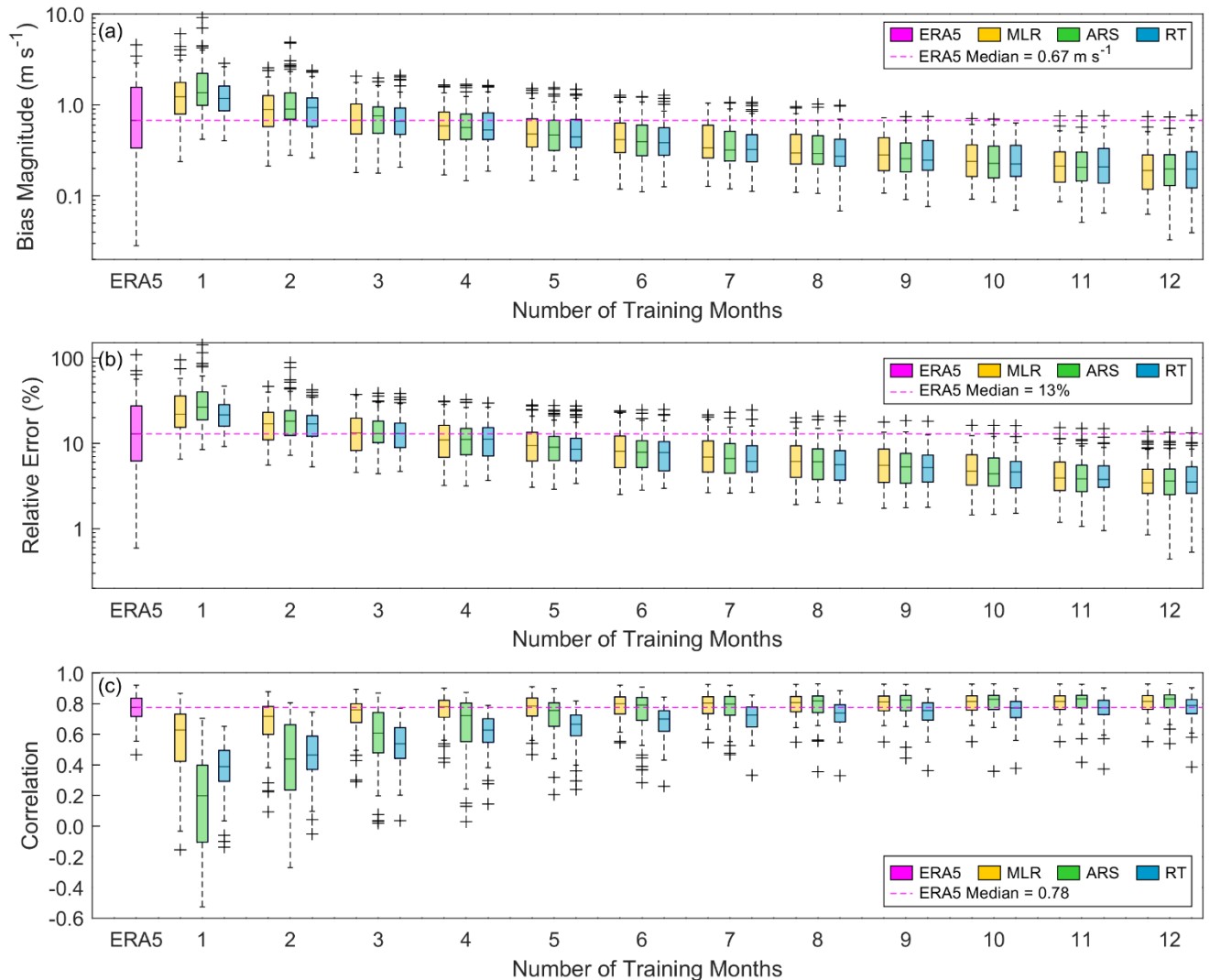

**Figure 7.** Worst case long-term (a) bias magnitude, (b) relative error, and (c) correlation for 66 sites comparing observations with ERA5 and MCP techniques using varying training period lengths.

To help us understand what those unfavourable conditions are that might lead to a worst-case scenario, we investigate the characteristics of the wind speed observations during the worst-case scenario periods relative to the average characteristics of the wind speed observations across all training periods with the same duration (Figure 8). It is interesting to note that the dates of the training time periods with the worst-case scenarios are variable according to MCP algorithm and error metric, particularly for the shortest duration training periods. We find that both the mean and standard deviation of the observed wind speeds during the worst-case scenario month or consecutive months tend to be lower than the mean and standard deviation of the observed wind speeds across all periods with the same durations, particularly when using ARS and RT

(Figure 8). In other words, low wind speed time periods with correspondingly low variations in the wind speeds tend to provide the biggest challenges for long-term MCP accuracy. For example, the median observed wind speed mean (standard deviation) across the 66 sites for all one-month duration periods is 5.80 m s$^{-1}$ (3.12 m s$^{-1}$), while the one-month worst-case scenario for ARS based on relative error corresponds to an observed wind speed mean (standard deviation) of 4.61 m s$^{-1}$ (2.30 m s$^{-1}$) (Figure 8a,b). Increasing numbers of training months correspond with higher mean wind speeds and standard deviations during the worst-case periods, along with convergence of the worst-case scenario mean wind speeds and standard deviations to mean wind speeds and standard deviations across all training periods of the same durations.

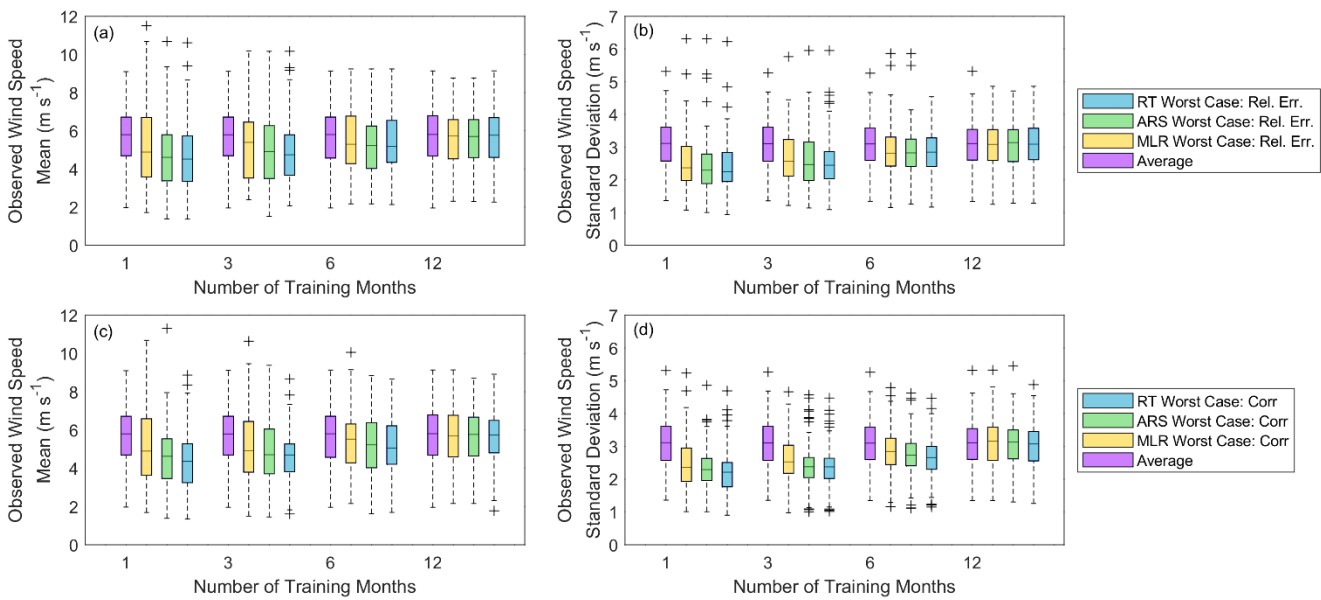

**Figure 8.** (a), (c) Mean wind speed and (b), (d) wind speed standard deviation across 66 sites for MLR, ARS, and RT worst-case scenarios for (a), (b) relative error and (c), (d) correlation, and across all training periods for each site of the same durations. Only the results using one, three, six, and twelve months of training are presented for the sake of brevity. Increasing numbers of training months correspond with convergence of the worst-case scenario mean wind speeds and standard deviations to mean wind speeds and standard deviations across all training periods of the same duration.

### 3.2 Regional long-term wind estimation performance

On average, using one month of observations produces regional median relative errors within 8% using any of the three MCP algorithms, except for the two Alaskan sites (Figure 9a-g). The Alaskan median relative errors are substantially higher (16% using MLR, 14% using ARS, and 15% using RT), but still significantly lower than the ERA5 median relative error for these two sites (28%) (Figure 3a). The most impressive reduction in bias magnitude occurs for the Southern Plains, where the ERA5 produces a median relative error of 20% (Figure 3a) that decreases to 3% (MLR) or 4% (ARS, RT) with the addition of just one month of observations (Figure 9e). Using three and six months of observations, all median regional relative errors are within 5% and 4%, respectively, except for Alaska where the median relative errors range from 11%

(ARS, RT) to 12% (MLR) using three months and 6% (ARS, RT) to 7% (MLR) using six months of observations. Using twelve months of observations, all regional bias magnitudes are within 3%, including Alaska (Figure 9a-g). A potential factor impacting the results for Alaska is the quality of the observations. While the automated quality-control techniques discussed in Section 2.1 remove periods of nonvarying wind speeds due to outages or icing, they may not capture more subtle impacts on the observations, such as partial icing of the anemometers.

On average, using RT with one month of observations degrades the ERA5 median correlations in all regions. Considering all sites, MLR tends to improve on the ERA5 correlations using one month of observations, while ARS performs similarly to ERA5 (Figure 5d). However, when exploring correlation on a regional scale, MLR performs similarly to ERA5 using one month of observations in all regions except the West, while ARS performs similarly to ERA5 in the Pacific Northwest, Midwest, and Northeast and disimproves relative to ERA5 at all other regions (Figure 9h-n).

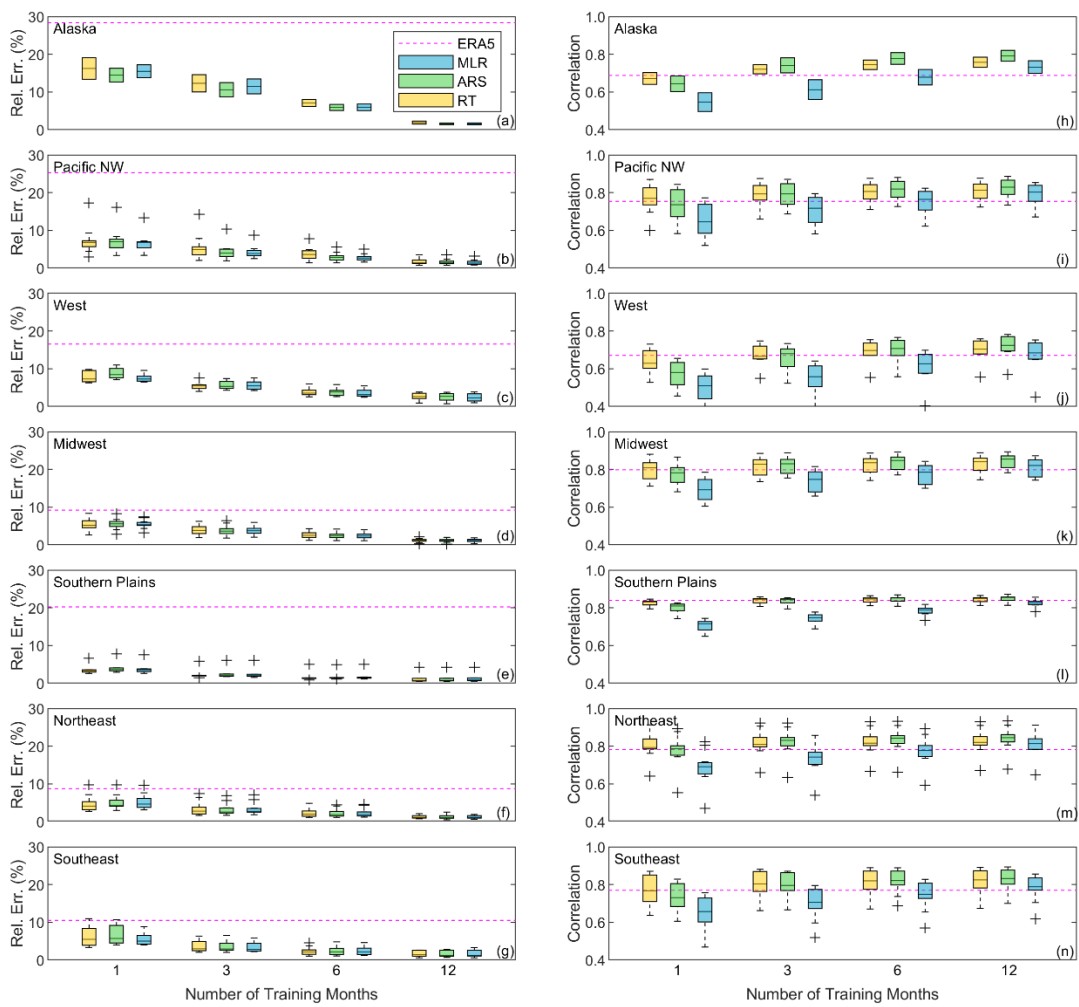

**Figure 9.** Long-term wind speed relative error (left) and correlation (right) according to region for varying training period lengths.

## 3.3 Seasonal relationships between observations and long-term wind estimation performance

When considering using less than one year of wind speed measurements to correct long-term wind speed estimates, it is important to select the most optimal time of year to gather such measurements (and to understand what times of year to avoid). For each of the 66 sites, Figure 10 and Figure 11 share the best and worst single months, on average, for optimizing error metrics when applying MCP to ERA5. The best months for optimizing relative error using all three MCP algorithms are quite variable across non-summer (June, July, August) months, with winter (December, January, February), spring (March, April, May), and fall (September, October, November) producing the most optimal relative errors at 21%-23%, 35%-41%, and 30%-33% of the 66 sites, respectively (Figure 10). For correlation, spring months produce the best results when using MLR and ARS for 55% and 48% of the sites, respectively. When using RT, fall months produce the best correlations at 41% of the sites. Summer months, particularly July and August, tend to produce the worst relative errors and correlations (Figure 11) and should be avoided if opting to create MCP-based wind speed estimates using a single season of observations. July and August produce the highest relative errors at 52%-56% of the 66 sites, depending on the selected algorithm, and the lowest correlations at 61%-73% of the sites.

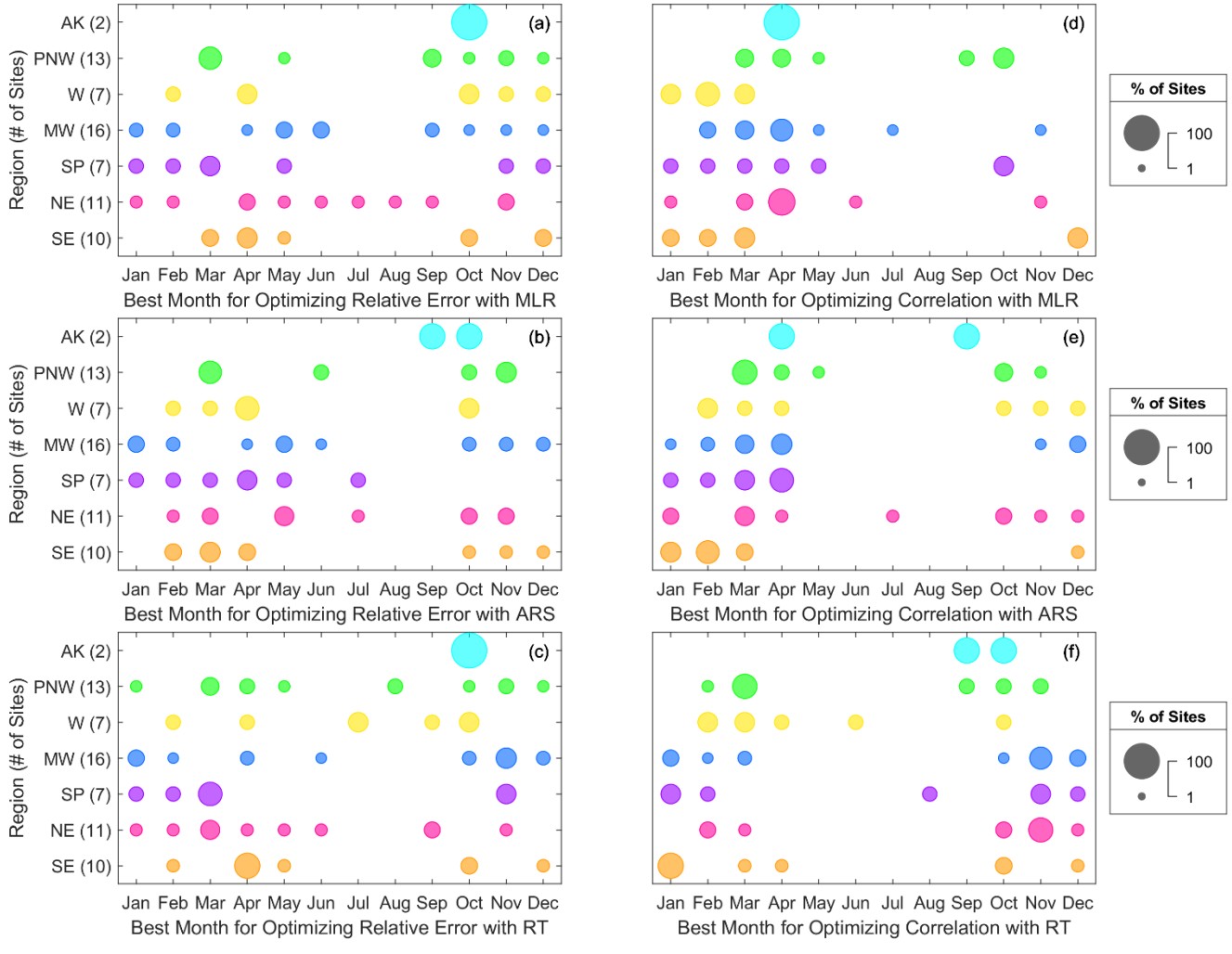

380

**Figure 10.** Best average single months for optimizing (a), (b), (c) relative error and (d), (e), (f) correlation when creating MCP-based long-term wind speed estimates using (a), (d) MLR, (b), (e) ARS, and (c), (f) RT according to US region. Larger circles indicate the best month for more sites in a given region.

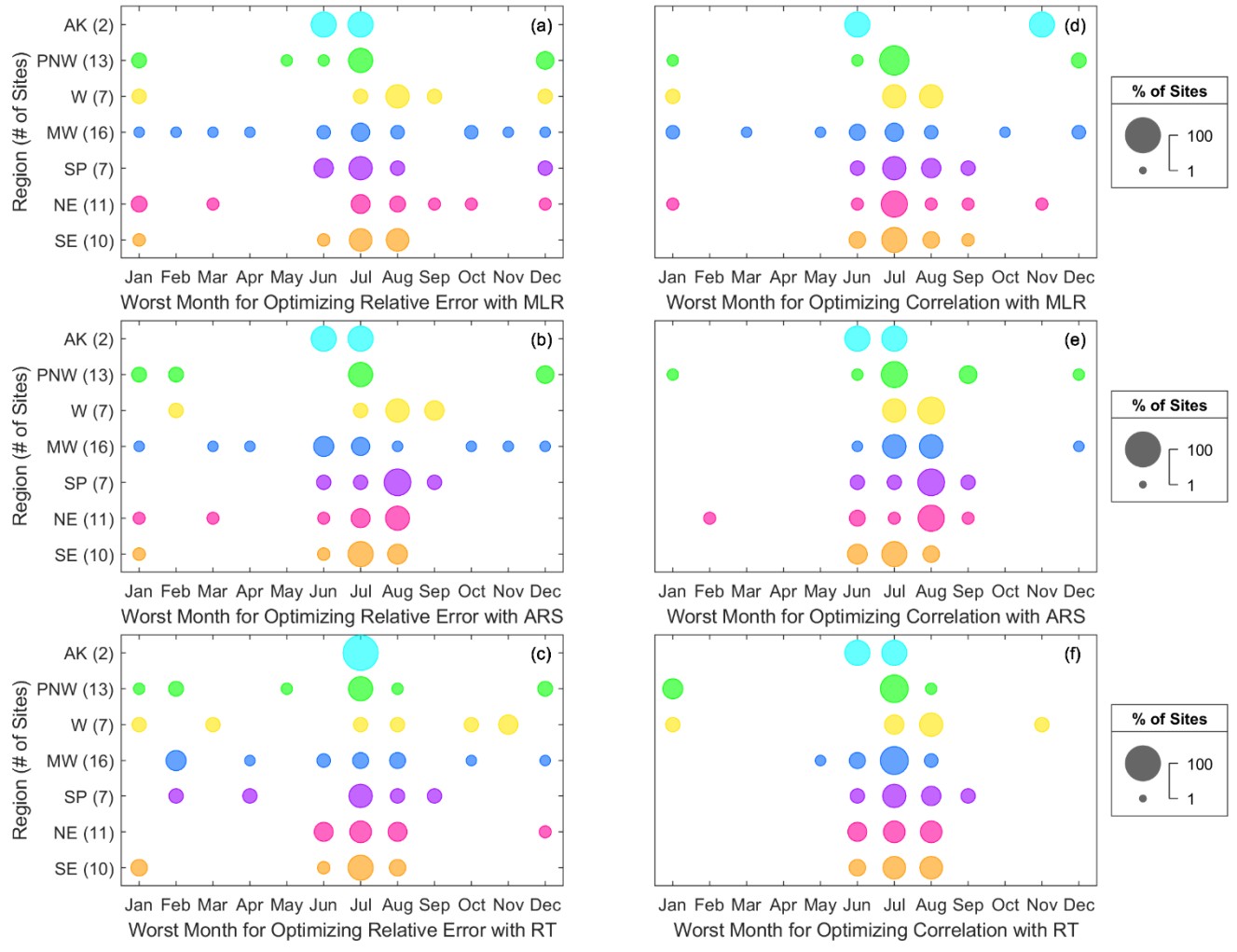

**Figure 11.** Worst average single months for optimizing (a), (b), (c) relative error and (d), (e), (f) correlation when creating MCP-based long-term wind speed estimates using (a), (d) MLR, (b), (e) ARS, and (c), (f) RT according to US region. Larger circles indicate the best month for more sites in a given region.

### 3.4 Implications for energy production estimates

On average, utilising months-long observations to correct reanalysis-based long-term wind speed estimates provides significant improvement in accuracy of predicted winds (Figure 5). However, substantial risks to accuracy occur when the months-long observational period is misrepresentative of longer-term wind speed trends (Figure 7). To evaluate the impacts of long-term correction using months-long observations on distributed wind energy production, we convert the long-term observed and simulated wind speeds to energy estimates using the National Renewable Energy Laboratory's 100-kW distributed wind reference power curve (NREL, 2019). Since variable lengths of long-term periods are utilised in this study,

we opt to consider energy production in terms of the capacity factor, i.e., the energy estimate divided by the product of the total number of hours in the long-term period and the turbine rated capacity (100 kW in this example). The observed wind speeds in this analysis produce gross (i.e., no loss considered) capacity factors ranging 3% (inland Louisiana) to 66% (Texas panhandle), with an average (median) of 34% (36%). The low wind speeds at some of these sites are not suitable for wind energy deployment. Additionally, the power production at the low wind sites will be dominated by the tail end of the wind speed distribution, leading to potential significant differences between the skill of the MCP algorithms in reproducing the highest percentiles of wind speeds versus estimating mean wind speeds, as in Sections 3.1-3.3. Therefore, sites with capacity factors based on observed wind speeds of at least 10% (58 sites in total) are considered for the following analysis.

Across the 58 observational sites, simply using ERA5 to produce long-term energy estimates results in capacity factor relative errors (|capacity factor based on simulations – capacity factor based on observed wind speeds| / capacity factor based on observed wind speeds) up to 89%, with a median of 24%. On average, just one month of observations reduces the capacity factor error range to within 51% and the median capacity factor relative error to within 10% (Figure 12a, Table 3). The largest ERA5-based outlier (89%) is for a complex terrain site in Oregon near the Columbia River with an observation-based capacity factor of 48%, and MCP correction using one month of onsite observations reduces the capacity factor relative error at this site to 8%-11% on average, depending on the algorithm. As for the wind speed relative errors (Figure 5b, Table 2), RT is the best algorithm on average for reducing capacity factor error when using the 100-kW reference power curve.

Like the wind speed evaluation, the potential for reduction of wind energy estimate error using months-long onsite observations comes with risk. With a one-month training period in the worst-case scenario, the ERA5 capacity factor error range (within 89%) increases to within 94%, and the median capacity factor errors across the 58 sites increase from 24% to 40%-47% (Figure 12b). Using three, six, and twelve months of training reduces the worst-case capacity factor errors to within 91% (median $\leq$ 26%), 82% (median $\leq$ 16%), and 37% (median $\leq$ 9%), respectively.

**Table 3.** Median site-average capacity factor relative errors according to algorithm and number of training months.

| Error Metric | Algorithm | Number of Training Months | | | | | | | | | | | |
|---|---|---|---|---|---|---|---|---|---|---|---|---|---|
| | | 1 | 2 | 3 | 4 | 5 | 6 | 7 | 8 | 9 | 10 | 11 | 12 |
| CF Relative | MLR | 9.7 | 8.5 | 7.6 | 6.8 | 6.2 | 5.6 | 4.9 | 4.3 | 4.1 | 3.7 | 3.5 | 3.5 |
| Error (%) | ARS | 9.9 | 8.0 | 7.3 | 6.7 | 6.2 | 5.6 | 5.0 | 4.6 | 4.4 | 4.2 | 3.9 | 3.8 |
| | RT | 8.6 | 6.8 | 5.9 | 5.0 | 4.3 | 4.0 | 3.7 | 3.4 | 3.2 | 3.0 | 3.0 | 3.1 |

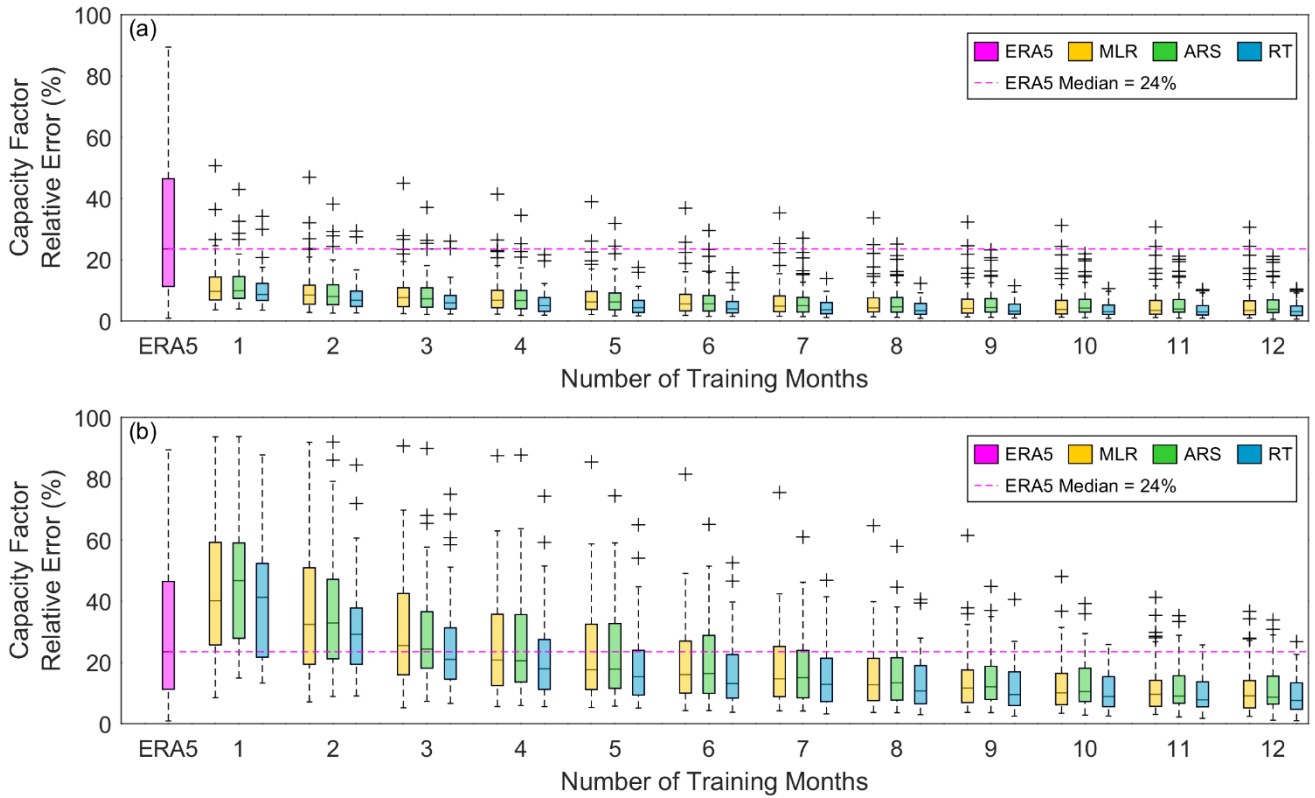

**Figure 12.** (a) Average and (b) worst-case scenario capacity factor relative error (|ERA5/MCP capacity factor – capacity factor based on observed wind speeds|/capacity factor based on observed wind speeds) according to number of training months for 58 sites with observation-simulated capacity factors of at least 10%.

## 3.5 Performance comparison with higher-resolution wind datasets

ERA5 has shown to be a valuable reference dataset for developing long-term wind speed estimates via MCP with short-term observations thanks to its extensive temporal coverage and relative success at representing fluctuations in observed wind speeds. However, ERA5 has the limitations of coarse horizontal resolution and a tendency to exhibit a slow wind speed bias (Ramon et al., 2019; Gualtieri, 2021; Sheridan et al., 2022; Wilczak et al., 2024), which urges comparison of the long-term MCP results with long-term estimates from higher-resolution wind datasets. We explore the performance of the MCP-based long-term wind speed estimates relative to Global Wind Atlas version 3 (GWA3) and the climatology component of the WIND Toolkit Long-term Ensemble Dataset (WTK-LED Climate), for which long-term wind speed estimates are freely and easily accessed through user friendly web applications.

GWA3 is produced by the Technical University of Denmark (DTU) and the World Bank Group. The developers used the Weather Research and Forecasting (WRF) mesoscale model (Skamarock et al., 2008) in conjunction with the Rapid Radiative Transfer Model (RRTM) for the longwave and shortwave radiation schemes (Mlawer et al., 1997; Iacono et al.,

2008), the Mellor-Yamada-Janjić planetary boundary layer (PBL) scheme (Janjić 1994), and ERA5 as the input and boundary conditions to produce simulated wind data at a horizontal resolution of 3 km (Davis et al., 2023). Next, microscale modelling was performed using the Wind Atlas Analysis and Application Program (WAsP) model (Troen and Petersen, 1989) with an output grid spacing of 250 m for GWA3. GWA3 provides global coverage for land-based wind estimates and offshore wind estimates within 200 km of shorelines. Long-term wind data are output at five heights between 10 m and 200 m based on the 10-year period of 2008-2017, and wind speed indices illustrate trends at annual, monthly, and diurnal temporal resolutions. Users can access GWA3 through its web application (DTU, 2024).

WTK-LED Climate was released in 2024 as the wind climatology component, developed by Argonne National Laboratory, of the WIND Toolkit Long-term Ensemble Dataset, a wind resource dataset led by the National Renewable Energy Laboratory. The climatology dataset uses an accelerated version of RRTM for general circulation models (RRTMG) for the radiation schemes, ERA5 for the initial and boundary conditions, and YSU for the PBL scheme (Draxl et al., 2024). WTK-LED Climate covers North America at a 4-km horizontal spatial resolution and 1-hr temporal resolution for the 20-year period of 2000-2020. Through the WindWatts web application (NREL, 2024), users can access WTK-LED Climate long-term average and monthly wind speeds at seven output heights between 30 m and 140 m.

Across the 66 observation sites, we extract the long-term average wind speed estimates from GWA3 and WTK-LED Climate at the surrounding output heights to the measurement heights and adjust them to the measurement height with the power law (Eq. 4 and 5). We determine the bias and relative error as in Eq. 2 and Eq. 3, keeping in mind that this comparison involves the different definitions of "long-term" that a user of the GWA3 and WindWatts web applications would experience (10 years for GWA3, 20 years for WTK-LED Climate and varying lengths for the observations as shown in Figure 2b). For our observation sites, the median bias magnitudes for the WTK-LED Climate and GWA3 long-term estimates are 0.71 m s$^{-1}$ and 0.36 m s$^{-1}$, respectively, and the median relative errors are 12.8% and 6.1%. On average, using even one month of wind speed observations to create long-term MCP-based wind speed estimates provides an improvement in accuracy over the long-term estimates provided by the higher-resolution WTK-LED Climate and GWA3 (Figure 13, Table 2).

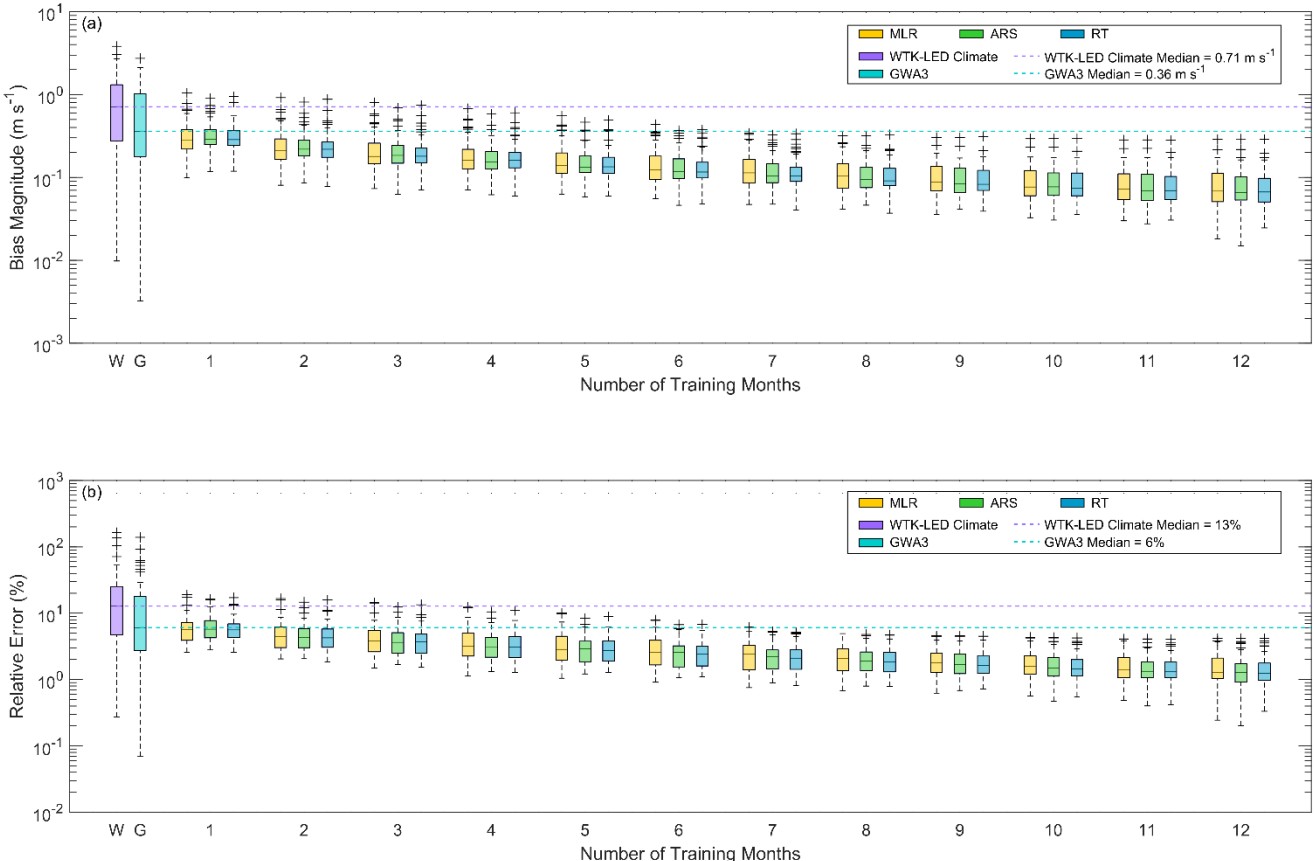

**Figure 13.** Average long-term wind speed (a) bias magnitude and (b) relative error across 66 sites based on the long-term estimates of GWA3 and WTK-LED Climate and the MCP-based estimates from ERA5 and short-term observations.

## 3.6 Recommendations and future work

It is important to consider the performance of wind estimation methodologies from a variety of statistical standpoints, since behind-the-meter customers may care most about whether their long-term production meets their initial expectations (bias) and front-of-the-meter customers may be also concerned with distribution network integration with other energy technologies (bias, correlation). While the three MCP algorithms assessed in this study perform similarly in terms of bias magnitude (Figure 5a), for the shortest training period lengths (one to three months) RT performs significantly worse on average for correlation than MLR and ARS (Figure 5d).

Given ERA5's popularity in the wind energy community, along with its known challenges for wind speed bias and relative success in terms of wind speed correlation, this study provides a useful framework for evaluating the performance of MCP-based corrections to long-term wind speed estimates using months-long observations. Additionally, the MCP estimates based on short-term observations are found to improve upon the long-term averages of recent wind datasets, including GWA3 and WTK-LED Climate. In the future, the exercise would benefit in from expansion to include additional long-term

models that provide wind resource information at different horizontal and vertical spatial resolutions. Additionally, the lessons learned in this work are being explored to quantify the geographic extent of their potential application, with aims to support broader wind speed bias correction in distributed wind tools and to reduce the amount of onsite measurement sites needed to correct sites with similar wind profile characteristics.

Based on this analysis, we identify the following key recommendations for wind energy developers creating long-term wind resource estimates under the constraint of less than one year of onsite measurements which may occur for any number of reasons, including instrument outages or timing of funding opportunities:

- While even one month of onsite wind speed measurements improves long-term wind speed estimates on average, incorporating at least four months of onsite measurements is a better option to mitigate the errors that could occur if some of the measured and reference wind speeds during the measurement period are poorly correlated.

- Summer months, particularly July and August, should be avoided if opting to create MCP-based wind speed estimates using a single season of measurements in the U.S., as these months tend to be the least representative of long-term wind speed means and standard deviations.

- Since potential wind customers may care about long-term wind resource accuracy from multiple viewpoints, it is important to note that of the three MCP algorithms explored, RT produces the lowest wind speed and capacity factor relative errors and ARS the lowest wind speed correlations. However, MLR is the least risky algorithm given the possibility of poor correlations between the measurements and the reference data.

The results of this work highlight the potential for anemometer or lidar loan programs to affordably assist future distributed wind energy customers with more accurate long-term wind resource estimates while maximizing the number of customers that can be served by reducing the measurement time needed.

**Code and data availability**

Many of the wind speed measurement datasets that support this study are publicly available. Measurements from U.S. DOE-sponsored laboratories, studies, and programs can be found at ANL (2022), ARM (2021), B2SHARE (2020), BNL (2020), LBNL (2020), NREL (2022). Measurements in the Pacific Northwest from the Bonneville Power Administration can be found at BPA (2022). Coastal measurements from the National Data Buoy Center are sourced from NDBC (2024). Measurements from the University of Vermont's Forest Ecosystem Monitoring Cooperative were formerly available at FEMC (2020). The remaining measurements are proprietary, subject to non-disclosure agreements, and have restricted access at DOE (2024).

The ERA5 reanalysis data is obtained from ECMWF (2024). The GWA3 wind data is available from DTU (2024). The WTK-LED Climate wind speed estimates were accessed through the WindWatts web application (NREL, 2024). The 100-kW reference distributed wind power curve utilised in this work is provided at NREL (2019). Data processing scripts are written in Matlab and are available from the author upon request.

**Author contributions**

Data management, software development, analysis, and manuscript preparation were performed by L. Sheridan. Team management was performed by H. Tinnesand and C. Phillips. All authors contributed to the research conceptualization, manuscript edits, and technical review.

**Competing interests**

The contact author has declared that none of the authors have any competing interests.

**Acknowledgements**

This work was authored by the Pacific Northwest National Laboratory, operated for the U.S. Department of Energy (DOE) by Battelle (contract no. DE-AC05-76RL01830). This work was authored in part by the National Renewable Energy Laboratory, operated by Alliance for Sustainable Energy, LLC, for the U.S. DOE under contract no. DE-AC36-08GO28308. Funding was provided by the U.S. DOE Office of Energy Efficiency and Renewable Energy Wind Energy Technologies

Office. The views expressed in the article do not necessarily represent the views of the DOE or the U.S. Government. The U.S. Government retains and the publisher, by accepting the article for publication, acknowledges that the U.S. Government retains a nonexclusive, paid-up, irrevocable, worldwide license to publish or reproduce the published form of this work, or allow others to do so, for U.S. Government purposes. The authors would like to thank Patrick Gilman and Bret Barker at the U.S. DOE Wind Energy Technologies Office for funding this research. The authors also wish to thank Alyssa Matthews and

three anonymous reviewers for their thoughtful suggestions for improvement of this work.

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
