# Peer review of "Evaluating the potential of short-term instrument deployment to improve distributed wind resource assessment"

_Wind Energy Science, 2024_

## Referee Comment (RC1)

Review of "Evaluating the potential of short-term instrument deployment to improve distributed wind resource assessment" by Lindsay M. Sheridan, Dmitry Duplyakin, Caleb Phillips, Heidi Tinnesand, Raj K. Rai, Julia E. Flaherty, and Larry K. Berg

**General comments**

The authors investigated the impact of the MCP correction on the long-term wind resource assessment. They focused on the accuracy of the MCP corrections using the months-long observations to investigate the possibility of reducing the measurement period used for the training data for the MCP. They found that one month of onsite wind speed measurements improves the long-term wind estimate on average, although four months of onsite measurements is a better option to mitigate the errors. It was also reported that the summer months should be avoided, as these months tend to be the least representative of long-term wind speed means and standard deviations. The study is well conducted, and the methods used are appropriate. The data are clearly presented. This study has shown quantitatively that the MCP, using a data period of less than one year, is effective in improving the performance of long-term wind resource assessment based on a large data set.

These findings will be of interest to wind energy developers working on the distributed wind resource assessment, as well as to researchers in the field. However, I have following concerns for the manuscript:

- Page 7, Line 131–136: Correlation coefficient, bias, and MAE were used as error metrics. In the case of bias and MAE, they would be associated with the magnitude of the values. Accordingly, the use of relative values would be more appropriate when comparing these results for different wind climates.

- Also, the accuracy of ERA 5 would depend on the measurement height as well as the region. I'm not so sure that the combined results can show the true performance of the ERA 5 dataset near the surface. If the accuracy is strongly dependent on the measurement height, it would be better to narrow the range of observations used for the analysis.

- Page 7, Lines 158–160: In addition to the three MCP algorithms used in this study, the other algorithms were also available. In fact, a commercial tool, such as WindPro provides methods using matrix and neutral network approaches. The reasons why these algorithms were chosen were briefly explained, but it is still unclear. Are there any reasons why they were chosen, e.g. because they gave better results than the other algorithms?

- Page 11, Figure 4: The box plots for each month in the figures are based on the different numbers of samples. Is it possible to add the number of samples used for each box plot on the right axis? The authors would analyze a large dataset to derive the results. The information of the sample size would make it easier for readers to understand how much data was used in the analysis.

- Page 22, Figure 12: Figure 12 (b) shows that the capacity factor errors appear to decrease when the training months reach four months. In the discussion, the authors concluded that four months is the preferred length of training months. If the aim of the investigation is to assess the capacity factor, is the MCP based the months-long observation an appropriate approach?
- The authors investigated the impact of observations using the error metrics with bias, MAE, and correlations. As shown in most of the figures in this study, the MCP methods would mainly affect the bias correction. Also, the improvement on the MAE scores would be due to the bias reduction, as discussed in Mattihas and Focken (2006). Is it necessary to use MAE and correlation coefficient for the KPIs as well as bias through the manuscript?
  - Lange, Matthias, and Ulrich Focken. *Physical approach to short-term wind power prediction*. Vol. 208. Berlin: Springer, 2006.

**Minor comments**
- Page 7, Figure 3: The error metrics were compared across seven regions. However, the number of sites used to drive the statistics would be different. One option would be to show the number of sites for each region in Figure 1.
- Page 7, Figure 3: The results of this study would be consistent with previous studies on the accuracy of the ERA5 dataset. The authors would be able to link the result to the previous study.
- Page 8, Line 216–221: It's difficult to follow the numbers described in the main text. Please consider using more tables to show the numbers.
- Page 15, Figure 6: Studying the worst-case scenario is certainly an interesting approach to investigating the risk of the MCP with the months-long observation. On the other hand, the large errors for the data with fewer training months would be due to the outliers. I assume that there is a possibility to improve such errors by applying robust regression algorithms that are insensitive to the outliers.

---

## Referee Comment (RC2)

**Reviewer's comments on "Evaluating the potential of short-term instrument deployment to improve distributed wind resource assessment" by Lindsay M. Sheridan, Dmitry Duplyakin, Caleb Phillips, Heidi Tinnesand, Raj K. Rai, Julia E. Flaherty, and Larry K. Berg**

The article investigates the errors associated with using wind measurements which are shorter than one year in measure correlate predict (MCP) methods to obtain representative long-term wind climates. The authors use a relatively large dataset comprising sites with a wide variation of wind conditions from different areas of the US. The results are presented in a clear manner and explained and discussed in detail. The findings highlight the value of using short-term measurements in combination with reanalysis data to improve wind resource estimations when compared to only relying on reanalysis data. The authors conclude that measurement data with durations as short as one month provide significant benefits but recommend using at least four months of measurement data.

The findings will be of interest for the wind energy community as mobile remote sensing devices like lidars have reduced the logistics associated with installing wind measurements (compared to mast-based measurements) and make short-term measurements much more viable. Moreover, short measurement periods are often used in the early stages of a measurement campaign to make intermediate assessments of the viability of wind energy projects.

However, the following general points need to be addressed before I can recommend the manuscript for publication:

- Section 2.1: The analysis is based on quite a large dataset. While presenting lengths and heights of the individual datasets, information on the observed mean wind speeds is missing. I strongly recommend including a histogram of the mean wind speeds or at least some statistics characterizing the mean wind speeds over all stations – such as average mean wind speed, standard deviation, minimum and maximum.

- Section 2.3: The authors use several different MCP methods. These include multiple linear regression, adaptive regression splines and regression trees. Linear regression but also other methods using a cost function optimizing the squared deviation between the model and the observations are well suited to perform bias corrections but have a strong tendency to create a negative bias in the variance. While the importance of errors in the variance of the long-term wind climate for resource estimation is usually smaller than the impact of errors in the mean wind speed it can be significant. For this reason, variance-conserving MCP methods have been developed [1, 2] and are now widely used in wind resource assessments. The authors should therefore clearly explain this limitation in the methods section and include the implications for estimations of annual energy production when discussing the results.

- Section 2.3: When introducing the MCP methods some important details remain unclear. The hyperparameters for the regression tree method are not specified. The authors should also explain how these hyperparameters were chosen. It remains unclear how the wind direction is used in the multiple linear regression approach. Due to its angular nature – i.e. 359° is next to 0° - the application of a linear regression approach including wind speed does not appear to be straightforward. In the industry, it is common to apply sectoral regression MCP [3] – i.e. binned by wind direction sectors. Authors should clearly explain why a different approach was chosen here and how their approach differs.

- Section 2.3 and section 3 and section 4: The presented analysis is mainly motivated by its relevance for resource assessments. However, out of the chosen error scores only the bias magnitude is of practical relevance for this application. While indicating the performance in reproducing temporal patterns, correlation and mean absolute errors are only of secondary importance in estimating AEP. This should be clearly addressed in section 2.3 and section 3 and section 4. While for other applications correlation and MAE might be more important, these applications are only briefly mentioned in lines 415ff. The provision of the standard deviation of the bias would be a useful additional performance measure as it corresponds to the uncertainty definition that is usually used in resource assessments.

- Section 2.3 and section 3: Wind conditions differ strongly between the different locations (cf. figure 7). The bias and MEA should therefore be presented in relative rather than absolute values or at least in relative values in addition to the absolute values currently given.

- Section 3.4: The approach chosen, and the conclusions drawn here are misleading for several reasons. Firstly, the analysis for all 6-months periods is performed for different sites. The different wind characteristics of these sites can cause differences in the performance of the MCP methods independently of the length of the long-term period. The observed differences might be caused by other reasons or just be coincidental. Instead of using different locations, locations with longer long-term period should be split-up artificially to obtain robust results. Secondly, increasing the length of the long-term period will result in more 6-months short-term periods in the analysis. This in turn will cause a worse performance in the worst-case scenario. This effect, however, is purely due to considerations in probability theory. A decline in the worst-case performance does not automatically relate to 'climate evolution' (line 362) as suggested. Comparing long-term periods with varying lengths directly will, thus, result in a distorted picture.

- Section 4: The conversion of the estimated long-term wind climates into energy provides significant added value for wind energy applications. However, the results should be presented using relative errors in the capacity factors rather than absolute values to make them more comparable. This is especially important since the reported capacity factors vary over more than one order of magnitude. Moreover, it is advisable to exclude locations with a very low wind resource, since these locations are not suitable for exploitation of the wind resource. In addition, the power production at these sites will be dominated by the tail end of the wind speed distribution and the skill of the MCP methods to reproduce the highest percentiles of wind speeds might differ significantly from their performance for a bias correction.

- Section 4 has the heading 'Discussion'. However, several new results are presented in this section. Moreover, several recommendations are drawn and observations are discussed in parts of section 3 (e.g., section 3.3 recommends that summer months should be avoided). I therefore recommend to integrate section 4 as a subsection into section 3 and rename Section 3 'Results and Discussion'.

- L278: The authors state: '... it is imperative to consider the worst-case scenario errors ...'. While I agree that the worst-case scenario provides useful information, the current presentation and discussion of the results will overinflate the perceived uncertainties associated with using short-term measurements for MCP by looking at extreme cases and outliers. The authors should therefore either use e.g. the 90th percentile of the observed errors rather than the worst-case scenario or clearly explain that the worst-case scenario is a very conservative approach and cannot directly be interpreted as an uncertainty.

**Specific comments:**

- L104f.: Many of the measurement heights are significantly lower than modern wind turbines. This should be highlighted and the limitations stemming from this point should be addressed in the discussion.

- L131ff.: This section is not related to the heading of the subsection (Reanalysis model for long-term correction). Consider moving it to a separate subsection.

- L206f.: '... provided each month in the training period meets the data recovery and quality threshold of 75%.' Are there any seasonal patterns in data recovery i.e. caused by icing in winter? This could influence the results.

- L442 states 'The results of this work highlight the benefits of anemometer or lidar loan programs'. The performed analysis, however, only highlights the benefits of short-term onsite measurements. Anemometer loan programs only provide one option to facilitate these.

- L52ff.: Here the authors discuss previous research that was conducted on MCP methods. Liléo et al. [4] published a comprehensive report that should be included in the discussed literature and might also be useful when discussing the results.

- L116ff.: The characteristics and performance of the ERA5 dataset are discussed. Recently Wilczak et al. [5] published an evaluation of ERA5 evaluating regional biases in ERA5 for different regions in the US. This reference would provide value here and in the discussion in section 3.2.

- L238f.: 'Using one month of training, MLR provides higher correlations (median = 0.79) than ERA5 (median = 0.78)' This difference seems rather small and maybe not even statistically significant. Should be rephrased.

**References**

[1] Anthony L. Rogers, John W. Rogers, and James F. Manwell. Comparison of the performance of four measure–correlate–predict algorithms. *Journal of Wind Engineering and Industrial Aerodynamics*, 93(3):243–264, 2005.

[2] S.M. Weekes and A.S. Tomlin. Data efficient measure-correlate-predict approaches to wind resource assessment for small-scale wind energy. *Renewable Energy*, 63:162–171, 2014.

[3] MEASNET. Evaluation of site-specific wind conditions, 2022.

[4] S. Liléo, E. Berge, O. Undheim, R. Klinkert, and R. E. Bredesen. Long-term correction of wind measurements-state-of-the-art, guidelines and future work. Technical Report Elforsk report 13:18, 2013.

[5] J. M. Wilczak, E. Akish, A. Capotondi, and G. P. Compo. Evaluation and bias correction of the era5 reanalysis over the united states for wind and solar energy applications. *Energies*, 17(7), 2024.

---

## Author Comment (AC1)

**Review #2**

The article investigates the errors associated with using wind measurements which are shorter than one year in measure correlate predict (MCP) methods to obtain representative long-term wind climates. The authors use a relatively large dataset comprising sites with a wide variation of wind conditions from different areas of the US. The results are presented in a clear manner and explained and discussed in detail. The findings highlight the value of using short-term measurements in combination with reanalysis data to improve wind resource estimations when compared to only relying on reanalysis data. The authors conclude that measurement data with durations as short as one month provide significant benefits but recommend using at least four months of measurement data.

The findings will be of interest for the wind energy community as mobile remote sensing devices like lidars have reduced the logistics associated with installing wind measurements (compared to mast-based measurements) and make short-term measurements much more viable. Moreover, short measurement periods are often used in the early stages of a measurement campaign to make intermediate assessments of the viability of wind energy projects.

Thank you very much for your helpful review! We appreciate and value your time and suggestions.

However, the following general points need to be addressed before I can recommend the manuscript for publication:

• Section 2.1: The analysis is based on quite a large dataset. While presenting lengths and heights of the individual datasets, information on the observed mean wind speeds is missing. I strongly recommend including a histogram of the mean wind speeds or at least some statistics characterizing the mean wind speeds over all stations – such as average mean wind speed, standard deviation, minimum and maximum.

A histogram of the mean wind speeds has been added to Figure 2 per your suggestion.

[Figure]

**Figure 1.** (a) Measurement heights, (b) long-term measurement availability, and (c) long-term measurement wind speeds for the sites evaluated for long-term performance based on months-long observations.

• Section 2.3: The authors use several different MCP methods. These include multiple linear regression, adaptive regression splines and regression trees. Linear regression but also other methods using a cost function optimizing the squared deviation between the model and the observations are well suited to perform bias corrections but have a strong tendency to create a negative bias in the variance. While the

importance of errors in the variance of the long-term wind climate for resource estimation is usually smaller than the impact of errors in the mean wind speed it can be significant. For this reason, variance-conserving MCP methods have been developed [1, 2] and are now widely used in wind resource assessments. The authors should therefore clearly explain this limitation in the methods section and include the implications for estimations of annual energy production when discussing the results.

To document the limitations of the algorithms utilized in this work per your suggestions, the following has been added to Lines 184-188: "Numerous additional algorithms have been developed and tested for their ability to improve simulation accuracy, and it is important to note that each feature different approaches, computational investments, complexities, skills, and limitations. For example, Rogers et al. (2005) note that linear regression techniques are easily implemented and well suited for performing bias correction but have a tendency to create a bias in the variance that variance-conserving MCP techniques are better suited to resolve."

• Section 2.3: When introducing the MCP methods some important details remain unclear. The hyperparameters for the regression tree method are not specified. The authors should also explain how these hyperparameters were chosen. It remains unclear how the wind direction is used in the multiple linear regression approach. Due to its angular nature – i.e. 359° is next to 0° - the application of a linear regression approach including wind speed does not appear to be straightforward. In the industry, it is common to apply sectoral regression MCP [3] – i.e. binned by wind direction sectors. Authors should clearly explain why a different approach was chosen here and how their approach differs.

Thank you for your suggestion to add more information concerning the hyperparameters. We have supplemented Lines 177-184 as follows:

"Adaptive regression splines involve the construction of piecewise-cubic regression models based on the short-term target and reference datasets (Jekabsons, 2016). In this analysis, we utilise the default parameter configurations of Jekabsons (2016). The maximum number of basis functions follows the formula of Milborrow (2016): min(200, max(20, 2*the number of input variables)) + 1. The maximum degree of interactions between input variables is set to 1 for additive modelling, therefore the generalized cross-validation penalty per knot is set to 2 following the recommendation of Friedman (1991). Regression trees recursively partition and evaluate the concurrent short-term target and reference datasets into unique segments, which are subsequently used to predict long-term target behaviour. In this analysis, the ensemble aggregation method used is least-squares boosting with 100 learning cycles."

Friedman, J. H.: Multivariate Adaptive Regression Splines (with discussion), The Annals of Statistics, Vol. 19, No. 1, 1991.

Jekabsons, G.: Adaptive Regressions Splines toolbox for Matlab/Octave, version 1.13.0, http://www.cs.rtu.lv/jekabsons/Files/ARESLab.pdf, 2016.

Milborrow, S.: Earth: Multivariate Adaptive Regression Spline Models [code] (derived from code by Hastie, T. and Tibshriani, R.), https://cran.r-project.org/web/packages/earth/index.html, 2016.

We agree that using the wind direction was a misguided approach and have taken the opportunity to rework the analysis using the u and v components instead. Thank you for this helpful suggestion!

• Section 2.3 and section 3 and section 4: The presented analysis is mainly motivated by its relevance for resource assessments. However, out of the chosen error scores only the bias magnitude is of practical relevance for this application. While indicating the performance in reproducing temporal patterns, correlation and mean absolute errors are only of secondary importance in estimating AEP. This should be clearly addressed in section 2.3 and section 3 and section 4. While for other applications correlation and

MAE might be more important, these applications are only briefly mentioned in lines 415ff. The provision of the standard deviation of the bias would be a useful additional performance measure as it corresponds to the uncertainty definition that is usually used in resource assessments.

Thank you for this suggestion. We have removed MAE as a featured error metric from the manuscript, though have kept correlation as we find it to be relevant for evaluating the performance of simulations in representing fluctuations in the wind, which is of interest when converting to power and assessing the implications of integration into a distribution network.

We have added the standard deviation to Figure 5 per your recommendation.

[Figure]

**Figure 2.** Average long-term (a) bias magnitude, (b) relative error, (c) standard deviation of bias magnitude, and (d) correlation for 66 sites comparing observations with ERA5 and MCP techniques using varying training period lengths, along with (e) the number of training samples per site and per number of training months.

• Section 2.3 and section 3: Wind conditions differ strongly between the different locations (cf. figure 7). The bias and MEA should therefore be presented in relative rather than absolute values or at least in relative values in addition to the absolute values currently given.

Per your helpful suggestion, we have added the relative error throughout the results section.

• Section 3.4: The approach chosen, and the conclusions drawn here are misleading for several reasons. Firstly, the analysis for all 6-months periods is performed for different sites. The different wind characteristics of these sites can cause differences in the performance of the MCP methods independently of the length of the long-term period. The observed differences might be caused by other reasons or just be coincidental. Instead of using different locations, locations with longer long-term period should be split-up artificially to obtain robust results. Secondly, increasing the length of the long-term period will result in more 6-months short-term periods in the analysis. This in turn will cause a worse performance in the worst-case scenario. This effect, however, is purely due to considerations in probability theory. A decline in the worst-case performance does not automatically relate to 'climate evolution' (line 362) as suggested. Comparing long-term periods with varying lengths directly will, thus, result in a distorted picture.

Thank you for pointing out the flaws in this analysis. We agree with your concerns and have removed Section 3.4 from the manuscript.

• Section 4: The conversion of the estimated long-term wind climates into energy provides significant added value for wind energy applications. However, the results should be presented using relative errors in the capacity factors rather than absolute values to make them more comparable. This is especially important since the reported capacity factors vary over more than one order of magnitude. Moreover, it is advisable to exclude locations with a very low wind resource, since these locations are not suitable for exploitation of the wind resource. In addition, the power production at these sites will be dominated by the tail end of the wind speed distribution and the skill of the MCP methods to reproduce the highest percentiles of wind speeds might differ significantly from their performance for a bias correction.

We appreciate your suggestion and have converted the capacity factor analysis to a study using relative errors and have removed the sites with very low wind resource using the threshold of capacity factors based on observed wind speeds that are < 10%.

[Figure]

**Figure 3.** (a) Average and (b) worst-case scenario capacity factor relative error (|ERA5/MCP capacity factor – capacity factor based on observed wind speeds|/capacity factor based on observed wind speeds) according to number of training months for 58 sites with observation-simulated capacity factors of at least 10%.

• Section 4 has the heading 'Discussion'. However, several new results are presented in this section. Moreover, several recommendations are drawn and observations are discussed in parts of section 3 (e.g., section 3.3 recommends that summer months should be avoided). I therefore recommend to integrate section 4 as a subsection into section 3 and rename Section 3 'Results and Discussion'.

Section 3 is now "Results and Discussion" with the former Section 4 integrated as Subsection 3.4, per your recommendation.

• L278: The authors state: '... it is imperative to consider the worst-case scenario errors ...'. While I agree that the worst-case scenario provides useful information, the current presentation and discussion of the results will overinflate the perceived uncertainties associated with using short-term measurements for MCP by looking at extreme cases and outliers. The authors should therefore either use e.g. the 90th percentile of the observed errors rather than the worst-case scenario or clearly explain that the worst-case scenario is a very conservative approach and cannot directly be interpreted as an uncertainty.

Thank you for your suggestion on how readers should interpret the worst-case scenario results. We have added the following text to Lines 303-306:

"It is important to keep in mind that the worst-case scenario error analysis is a conservative approach that is not analogous to assessing algorithm uncertainty. Additionally, more robust algorithms than those studied in this work could reduce the sensitivity to the outliers in the shortest training timeseries that drive error in the long-term estimates."

**Specific comments:**

• L104f.: Many of the measurement heights are significantly lower than modern wind turbines. This should be highlighted and the limitations stemming from this point should be addressed in the discussion.

While the findings of the manuscript are hoped to be of interest to multiple wind energy sectors, the work was funded to be of primary benefit to the distributed wind sector. Hub heights for distributed wind turbines are wide ranging and, for small distributed wind turbines (≤ 100 kW capacity), often occur at those lower heights between 20 m and 40 m.

We have modified Lines 105-107 to improve the relevance as follows: "Many of the lowest observations, which align with small distributed wind turbine hub heights (between 20 m and 40 m), source from the National Data Buoy Center and are located along coastlines. The highest observations, which align with large distributed wind turbine hub heights (between 80 m and 100 m), are in Long Island, New York (85 m) and the San Francisco Bay Area, California (100 m)."

Additionally, per a suggestion from another reviewer, we have provided the ERA5 error metrics broken out by height ranges to Figure 3 to provide more insight into model performance at various heights. We have also added the accompanying text to Lines 155-158: "No consistent trends in ERA5 performance are noted according to height above ground (Figure 3d, e, f). The wind speed relative errors are greatest for measurement heights between 30 m and 40 m (median = 31%), while the median relative errors for measurement heights between 1) 20 m and 30 m and 2) 40 m and 50 m are 11% and 10%, respectively."

[Figure]

**Figure 4.** Long-term ERA5 wind speed (a), (d) bias (b), (e) relative error, and (c), (f) correlation across 66 measurement sites in the United States, grouped by region (top) and measurement height (bottom). AK = Alaska, PNW = Pacific Northwest, W = West, MW = Midwest, SP = Southern Plains, NE = Northeast, and SE = Southeast.

• L131ff.: This section is not related to the heading of the subsection (Reanalysis model for longterm correction). Consider moving it to a separate subsection.

Agreed. We have moved it to a separate subsection (2.3) entitled "Metrics for performance evaluation."

• L206f.: '... provided each month in the training period meets the data recovery and quality threshold of 75%.' Are there any seasonal patterns in data recovery i.e. caused by icing in winter? This could influence the results.

We appreciate this interesting suggestion, and have added a new figure and accompanying text to the manuscript in response (Lines 233-235):

"Across the measurement sites, calendar months in the spring and fall had the most single instances of ≥75% data recovery and quality, followed by summer, and lastly winter. Median measurement data recovery and quality percentages according to calendar month ranged from 99.2% (December) to 99.7% (May)."

[Figure]

• L442 states 'The results of this work highlight the benefits of anemometer or lidar loan programs'. The performed analysis, however, only highlights the benefits of short-term onsite measurements. Anemometer loan programs only provide one option to facilitate these.

The sentence has been rephrased as follows (Lines 450-452): "The results of this work highlight the potential for anemometer or lidar loan programs to affordably assist future distributed wind energy customers with more accurate long-term wind resource estimates while maximizing the number of customers that can be served by reducing the measurement time needed."

• L52ff.: Here the authors discuss previous research that was conducted on MCP methods. Lil´eo et al. [4] published a comprehensive report that should be included in the discussed literature and might also be useful when discussing the results.

Thank you for this reference. We have added it to the discussion on Lines 53-55: "The vast majority of wind resource assessment literature supports collecting at least one year of onsite measurements to represent a full seasonal wind cycle, including the analyses of Dinler (2013), Liléo et al. (2013), Mifsud et al. (2018), Zakaria et al. (2018), Tang et al. (2019), and Chen et al. (2022)."

• L116ff.: The characteristics and performance of the ERA5 dataset are discussed. Recently Wilczak et al. [5] published an evaluation of ERA5 evaluating regional biases in ERA5 for different regions in the US. This reference would provide value here and in the discussion in section 3.2.

We have added the results from this helpful paper to our manuscript as follows:

Lines 130-132: "Using measurements from more than 100 onshore and offshore lidars, sodars, and meteorological towers across the United States, Wilczak et al. (2024) determined that ERA5-derived wind power estimates were biased low by 20%."

Lines 159-162: "The tendencies of ERA5 to underestimate the observed wind speeds in this analysis while exhibiting a relatively high degree of correlation with them aligns with the findings of Ramon et al.

(2019), Murcia et al. (2022), Sheridan et al. (2022), and Wilczak et al. (2024) discussed in Section 2.2. The bias trends according to region (Figure 3a) also align with the findings of Wilczak et al. (2024) in that ERA5 underestimation is noted in the Pacific Northwest and Southern Plains, while a mix of overestimation and underestimated is noted for the Midwest."

• L238f.: 'Using one month of training, MLR provides higher correlations (median = 0.79) than ERA5 (median = 0.78)' This difference seems rather small and maybe not even statistically significant. Should be rephrased.

We agree and have rephrased the sentence as follows (Lines 263-265): "Using one month of training, MLR and ARS produce similar correlations (medians = 0.79 and 0.78, respectively) to ERA5 (median = 0.78), while the RT correlations are quite a bit worse (median = 0.68)."

**References**

[1] Anthony L. Rogers, John W. Rogers, and James F. Manwell. Comparison of the performance of four measure–correlate–predict algorithms. Journal of Wind Engineering and Industrial Aerodynamics, 93(3):243–264, 2005.

[2] S.M.Weekes and A.S. Tomlin. Data efficient measure-correlate-predict approaches to wind resource assessment for small-scale wind energy. Renewable Energy, 63:162–171, 2014.

[3] MEASNET. Evaluation of site-specific wind conditions, 2022.

[4] S. Lil´eo, E. Berge, O. Undheim, R. Klinkert, and R. E. Bredesen. Long-term correction of wind measurements-state-of-the-art, guidelines and future work. Technical Report Elforsk report 13:18, 2013.

[5] J. M. Wilczak, E. Akish, A. Capotondi, and G. P. Compo. Evaluation and bias correction of the era5 reanalysis over the united states for wind and solar energy applications. Energies, 17(7), 2024.

---

## Author Comment (AC2)

**Review #1**

**General comments**

The authors investigated the impact of the MCP correction on the long-term wind resource assessment. They focused on the accuracy of the MCP corrections using the months-long observations to investigate the possibility of reducing the measurement period used for the training data for the MCP. They found that one month of onsite wind speed measurements improves the longterm wind estimate on average, although four months of onsite measurements is a better option to mitigate the errors. It was also reported that the summer months should be avoided, as these months tend to be the least representative of long-term wind speed means and standard deviations. The study is well conducted, and the methods used are appropriate. The data are clearly presented. This study has shown quantitatively that the MCP, using a data period of less than one year, is effective in improving the performance of long-term wind resource assessment based on a large data set.

Thank you for your time and your thoughtful review. We are very grateful for your feedback and support!

These findings will be of interest to wind energy developers working on the distributed wind resource assessment, as well as to researchers in the field. However, I have following concerns for the manuscript:

• Page 7, Line 131–136: Correlation coefficient, bias, and MAE were used as error metrics. In the case of bias and MAE, they would be associated with the magnitude of the values. Accordingly, the use of relative values would be more appropriate when comparing these results for different wind climates.

We appreciate this suggestion and have updated the results section to include relative error per your recommendation.

• Also, the accuracy of ERA 5 would depend on the measurement height as well as the region. I'm not so sure that the combined results can show the true performance of the ERA 5 dataset near the surface. If the accuracy is strongly dependent on the measurement height, it would be better to narrow the range of observations used for the analysis.

In response to your helpful question, we have added the error metrics according to measurement height to Figure 3. We have also added the accompanying text to Lines 155-158: "No consistent trends in ERA5 performance are noted according to height above ground (Figure 3d, e, f). The wind speed relative errors are greatest for measurement heights between 30 m and 40 m (median = 31%), while the median relative errors for measurement heights between 1) 20 m and 30 m and 2) 40 m and 50 m are 11% and 10%, respectively."

[Figure]

**Figure 1.** Long-term ERA5 wind speed (a), (d) bias (b), (e) relative error, and (c), (f) correlation across 66 measurement sites in the United States, grouped by region (top) and measurement height (bottom). AK = Alaska, PNW = Pacific Northwest, W = West, MW = Midwest, SP = Southern Plains, NE = Northeast, and SE = Southeast.

• Page 7, Lines 158–160: In addition to the three MCP algorithms used in this study, the other algorithms were also available. In fact, a commercial tool, such as WindPro provides methods using matrix and neutral network approaches. The reasons why these algorithms were chosen were briefly explained, but it is still unclear. Are there any reasons why they were chosen, e.g. because they gave better results than the other algorithms?

We agree that the three algorithms chosen are but a small subset of the techniques available for MCP exploration. Given the vast quantity of techniques available, we limited ourselves to three in order to optimize the effort we could spend analyzing their performance. In particular, we selected multiple linear regression and regression trees as they proved successful in our previous distributed wind-focused studies (Phillips et al., 2022) and added adaptive regression splines as a third method for comparison.

Phillips, C., Sheridan, L. M., Conry, P., Fytanidis, D. K., Duplyakin, D., Zisman, S., Duboc, N., Nelson, M., Kotamarthi, R., Linn, R., Broersma, M., Spijkerboer, T., and Tinnesand, H.: Evaluation of obstacle modelling approaches for resource assessment and small wind turbine siting: case study in the northern Netherlands, Wind Energ. Sci., 7, 1153–1169, https://doi.org/10.5194/wes-7-1153-2022, 2022.

• Page 11, Figure 4: The box plots for each month in the figures are based on the different numbers of samples. Is it possible to add the number of samples used for each box plot on the right axis? The authors would analyze a large dataset to derive the results. The information of the sample size would make it easier for readers to understand how much data was used in the analysis.

The box plots for each month are based on the average error metric for each site, so the boxes are composed of 66 values regardless of the number of months. But you are of course correct that different numbers of samples are going into the calculation of each of those averages according to number of training months. To help the readers understand how much data was used, we have added an additional

box plot to Figure 5 that shows the number of samples per site incorporated according to the number of training months.

[Figure]

**Figure 2.** Average long-term (a) bias magnitude, (b) MAE, and (c) correlation for 66 sites comparing observations with ERA5 and MCP techniques using varying training period lengths, along with (d) the number of training samples per site and per number of training months.

• Page 22, Figure 12: Figure 12 (b) shows that the capacity factor errors appear to decrease when the training months reach four months. In the discussion, the authors concluded that four months is the preferred length of training months. If the aim of the investigation is to assess the capacity factor, is the MCP based the months-long observation an appropriate approach?

We stress in the discussion that four months is the minimum amount of time, as opposed to the preferred, that observations should be gathered in order to mitigate the errors that could occur if some of the wind speeds in the measurement period are misrepresentative of the longer-term trends (Lines 440-442): "While even one month of onsite wind speed measurements improves long-term wind speed estimates on average, incorporating at least four months of onsite measurements is a better option to

mitigate the errors that could occur if some of the wind speeds in the measurement period are misrepresentative of the longer-term trends."

• The authors investigated the impact of observations using the error metrics with bias, MAE, and correlations. As shown in most of the figures in this study, the MCP methods would mainly affect the bias correction. Also, the improvement on the MAE scores would be due to the bias reduction, as discussed in Mattihas and Focken (2006). Is it necessary to use MAE and correlation coefficient for the KPIs as well as bias through the manuscript?

> Lange, Matthias, and Ulrich Focken. Physical approach to short-term wind power prediction. Vol. 208. Berlin: Springer, 2006.

We agree that some of our original choices of error metrics were redundant. We have removed MAE from the evaluation, but have kept correlation as we find it to be relevant for evaluating the performance of simulations in representing fluctuations in the wind, which is of interest when converting to power and assessing the implications of integration into a distribution network.

**Minor comments**

• Page 7, Figure 3: The error metrics were compared across seven regions. However, the number of sites used to drive the statistics would be different. One option would be to show the number of sites for each region in Figure 1.

We have updated Figure 1 with the number of observational sites in each region, per your helpful suggestion. Thank you!

[Figure]

**Figure 3.** Locations of wind measurements assessed for establishing long-term performance based on months-long observations used in this study. The number of observational sites per region is included in parentheses.

• Page 7, Figure 3: The results of this study would be consistent with previous studies on the accuracy of the ERA5 dataset. The authors would be able to link the result to the previous study.

Thank you for this suggestion to link the ERA5 performance from our analysis to previous work. We have added the following on Lines 159-164: "The tendencies of ERA5 to underestimate the observed wind speeds in this analysis while exhibiting a relatively high degree of correlation with them aligns with the findings of Ramon et al. (2019), Murcia et al. (2022), Sheridan et al. (2022), and Wilczak et al. (2024) discussed in Section 2.2. The bias trends according to region (Figure 3a) also align with the findings of Wilczak et al. (2024) in that ERA5 underestimation is noted in the Pacific Northwest and Southern Plains, while a mix of overestimation and underestimated is noted for the Midwest."

• Page 8, Line 216–221: It's difficult to follow the numbers described in the main text. Please consider using more tables to show the numbers.

We agree and have added Table 2 and Table 3 to provide easier reference for the discussion.

**Table 1.** Median biases, bias magnitudes, relative errors, and correlations according to algorithm and number of training months.

| Error Metric | Algorithm | Number of Training Months | | | | | | | | | | | |
| | | 1 | 2 | 3 | 4 | 5 | 6 | 7 | 8 | 9 | 10 | 11 | 12 |
|---|---|---|---|---|---|---|---|---|---|---|---|---|---|
| Bias (m s$^{-1}$) | MLR | -0.03 | -0.02 | -0.02 | -0.01 | 0.00 | 0.00 | 0.00 | 0.01 | 0.01 | 0.01 | 0.01 | 0.01 |
| | ARS | -0.04 | -0.02 | -0.02 | -0.01 | 0.00 | 0.00 | 0.01 | 0.01 | 0.01 | 0.01 | 0.01 | 0.01 |
| | RT | -0.08 | -0.04 | -0.03 | -0.01 | -0.01 | 0.00 | 0.00 | 0.01 | 0.00 | 0.00 | 0.00 | 0.00 |
| Bias Magnitude (m s$^{-1}$) | MLR | 0.28 | 0.21 | 0.18 | 0.16 | 0.14 | 0.12 | 0.11 | 0.10 | 0.09 | 0.08 | 0.07 | 0.07 |
| | ARS | 0.29 | 0.22 | 0.18 | 0.15 | 0.13 | 0.12 | 0.10 | 0.09 | 0.08 | 0.08 | 0.07 | 0.07 |
| | RT | 0.29 | 0.22 | 0.18 | 0.16 | 0.13 | 0.12 | 0.10 | 0.09 | 0.08 | 0.07 | 0.07 | 0.07 |
| Standard Dev. of Bias Magnitude (m s$^{-1}$) | MLR | 0.24 | 0.18 | 0.15 | 0.13 | 0.11 | 0.09 | 0.08 | 0.07 | 0.06 | 0.06 | 0.05 | 0.05 |
| | ARS | 0.28 | 0.19 | 0.15 | 0.12 | 0.10 | 0.08 | 0.07 | 0.07 | 0.06 | 0.05 | 0.05 | 0.05 |
| | RT | 0.24 | 0.18 | 0.14 | 0.12 | 0.10 | 0.08 | 0.07 | 0.06 | 0.06 | 0.05 | 0.05 | 0.05 |
| Relative Error (%) | MLR | 5.7 | 4.5 | 3.8 | 3.2 | 2.8 | 2.6 | 2.4 | 2.1 | 1.8 | 1.6 | 1.4 | 1.3 |
| | ARS | 5.7 | 4.4 | 3.6 | 3.1 | 2.9 | 2.6 | 2.2 | 1.9 | 1.7 | 1.5 | 1.3 | 1.3 |
| | RT | 5.6 | 4.3 | 3.7 | 3.1 | 2.8 | 2.4 | 2.1 | 1.8 | 1.6 | 1.4 | 1.3 | 1.2 |
| Correlation | MLR | 0.79 | 0.80 | 0.81 | 0.81 | 0.81 | 0.81 | 0.81 | 0.82 | 0.82 | 0.82 | 0.82 | 0.82 |
| | ARS | 0.77 | 0.80 | 0.81 | 0.82 | 0.83 | 0.83 | 0.83 | 0.83 | 0.83 | 0.84 | 0.84 | 0.84 |
| | RT | 0.67 | 0.70 | 0.72 | 0.74 | 0.76 | 0.77 | 0.78 | 0.78 | 0.79 | 0.80 | 0.80 | 0.80 |

**Table 2.** Median site-average capacity factor relative errors according to algorithm and number of training months.

| Error Metric | Algorithm | Number of Training Months | | | | | | | | | | | |
| | | 1 | 2 | 3 | 4 | 5 | 6 | 7 | 8 | 9 | 10 | 11 | 12 |
|---|---|---|---|---|---|---|---|---|---|---|---|---|---|
| CF Relative Error (%) | MLR | 9.7 | 8.5 | 7.6 | 6.8 | 6.2 | 5.6 | 4.9 | 4.3 | 4.1 | 3.7 | 3.5 | 3.5 |
| | ARS | 9.9 | 8.0 | 7.3 | 6.7 | 6.2 | 5.6 | 5.0 | 4.6 | 4.4 | 4.2 | 3.9 | 3.8 |
| | RT | 8.6 | 6.8 | 5.9 | 5.0 | 4.3 | 4.0 | 3.7 | 3.4 | 3.2 | 3.0 | 3.0 | 3.1 |

• Page 15, Figure 6: Studying the worst-case scenario is certainly an interesting approach to investigating the risk of the MCP with the months-long observation. On the other hand, the large errors for the data

with fewer training months would be due to the outliers. I assume that there is a possibility to improve such errors by applying robust regression algorithms that are insensitive to the outliers.

Thank you for pointing out this consideration. We have added the following text to Lines 303-306:

"It is important to keep in mind that the worst-case scenario error analysis is a conservative approach that is not analogous to assessing algorithm uncertainty. Additionally, more robust algorithms than those studied in this work could reduce the sensitivity to the outliers in the shortest training timeseries that drive error in the long-term estimates."

---

## Referee Report (RR1)

This manuscript investigates how short measurements could be long-term corrected (LTC) with reference data from ERA5 in order to obtain accurate estimates of long-term mean wind speed and energy production. The motivation is that full-scale (1y+) measurement campaigns are costly and often unfeasible for small-scale wind power projects.

Overall, the paper is clear and the study is robust. The dataset of wind speed measurements is relatively large and representative for small-scale wind power in the US. The manuscript addresses a real problem, which is well described, and is thus relevant. A nice aspect is the focus on worst case scenarios in addition to mean statistics.

There are however some possible improvements to methodology and text. Below are comments listed approximately in order of importance.

1. ERA5 is state-of-the-art for LTC and as input to mesoscale models. Due to e.g. its coarse resolution it is however not suitable for mean wind estimates at particular sites. Raw ERA5 is thus a poor baseline and all comparisons becomes more or less irrelevant. It would strengthen the paper greatly if e.g. GWA (global wind atlas, contains mean wind speed and parameters needed for energy calculations) was used as benchmark.

2. Overall, the paper is well written and methods are clear. The paragraphs on MCP and quality threshold (L197-207 + L223-235) are however difficult to follow in detail. As an example, it is not fully clear weather the authors exclude MCP-simulated wind speeds for periods with missing observations when computing means. A shorter, clearer and stepwise description would be better. From my understanding

    a. The mean of the full observation period is taken as ground truth.

    b. A model is trained on a short (few months) period

    c. The model is used to predict the full observation period.

    d. Mean statistics are computed and compared to ground truth. Samples with missing observations are excluded from the calculations.

3. One of the main reasons why MCP-methods will work well or not on short measurement series is the seasonality of the errors in the reference series. It is not a problem per se if the measurement period is not representative for the long-term as long as the reference series accurately captures the variations. As an extreme example, with perfectly correlated observation and reference series, one observation sample is enough to estimate the true long-term mean wind speed. In practice, as is well demonstrated in the manuscript, the risk of getting poor results are higher if measurements are taken during e.g. low-wind periods. But these phenomena should not be confused (as in e.g. L24-27, L440-442).

4. Energy results should really be in the results section (not in discussion). Since energy production is more important than mean wind speed for the intended application, it would be good to emphasize these results more (on the expense of mean wind speed), e.g. in discussion and abstract.

5. It would be good to include descriptions and references of state-of-the-art MCP methods.

6. Many different regression tree algorithms exist. State which one you use and give a reference.

7. The quality control of measurements is performed automatically. This is fine, in particular given the large number of observations. Periods with disturbed observations might however persist after the control, an example is partial icing of the anemometers. This could maybe explain the poorer results in Alaska.

8. L33-34: normally, one describes it as that reference data (e.g. ERA5) are used to correct a short measurement, not the other way around.

9. L58-65 could be rewritten to sound less like a sales text for ArcVera.

10. Refs to fig 5c should be 5d (L357 + L428)

11. L398 "versus performing a bias correction" could be "versus estimating mean wind speed" (the term bias correction should not be reserved for mean wind speed estimates).

---

## Referee Report (RR2)

**Reviewer's comments on revisions of "Evaluating the potential of short-term instrument deployment to improve distributed wind resource assessment" by Lindsay M. Sheridan, Dmitry Duplyakin, Caleb Phillips, Heidi Tinnesand, Raj K. Rai, Julia E. Flaherty, and Larry K. Berg**

The authors have thoroughly and appropriately addressed the reviewer's comments. I therefore recommend the manuscript for publication and only suggest one smaller technical correction:

- Section 3.4: The heading currently reads "Discussion". I suggest renaming this section for two reasons. First, the title "Discussion" suggests that the discussion of the results is limited to 3.4. However, results are evaluated and discussed in other parts of section 3 as well. Second, new results are presented in subsection 3.4 (e.g. table 3 and figure 12). A new heading should refer to the topic of the sub-section – i.e. "Implications for Capacity Factors/Annual Energy" production or similar.

---

## Referee Report (RR3)

**Evaluating the potential of short-term instrument deployment to improve distributed wind resource assessment** by Sheridan et al

The authors have adressed all my comments and I recommend to accept the paper. Two minor and optional suggestions:

- Make a reference to Section 3.5 in Section 3.1 (or maybe better; merge results for ERA5 and GWA/WTK, i.e. show GWA/WTK results in same figures as ERA5 in 3.1 + move description of these to Section 2)

- Better to cite the original algorithm (random forest, gradient boosting etc.) rather than Matlab implementation (although a bit difficult to understand from a quick look, maybe Matlab has its own algorithms?)

---

## Author Response (AR2)

**Reviewer 1**

This manuscript investigates how short measurements could be long-term corrected (LTC) with reference data from ERA5 in order to obtain accurate estimates of long-term mean wind speed and energy production. The motivation is that full-scale (1y+) measurement campaigns are costly and often unfeasible for small-scale wind power projects.

Overall, the paper is clear and the study is robust. The dataset of wind speed measurements is relatively large and representative for small-scale wind power in the US. The manuscript addresses a real problem, which is well described, and is thus relevant. A nice aspect is the focus on worst case scenarios in addition to mean statistics.

There are however some possible improvements to methodology and text. Below are comments listed approximately in order of importance.

**We really appreciate you taking the time to review our work and provide great suggestions for improvement. We have addressed and incorporated your ideas and edits as follows. Again, thank you!**

1. ERA5 is state-of-the-art for LTC and as input to mesoscale models. Due to e.g. its coarse resolution it is however not suitable for mean wind estimates at particular sites. Raw ERA5 is thus a poor baseline and all comparisons becomes more or less irrelevant. It would strengthen the paper greatly if e.g. GWA (global wind atlas, contains mean wind speed and parameters needed for energy calculations) was used as benchmark.

We really appreciate this recommendation for improving the baseline for comparison. To address your helpful suggestion, we have added Section 3.5 and Figure 13, which compare the performance of the MCP-based estimates with the long-term estimates from two higher-resolution wind datasets: 1) your suggestion of Global Wind Atlas (250 m resolution) and 2) the recently released wind climatology component of the WIND Toolkit Long-term Ensemble Dataset (4 km resolution).

"3.5 Performance comparison with higher-resolution wind datasets

[revised manuscript text omitted]

2. Overall, the paper is well written and methods are clear. The paragraphs on MCP and quality threshold (L197-207 + L223-235) are however difficult to follow in detail. As an example, it is not fully clear weather the authors exclude MCP-simulated wind speeds for periods with missing observations when computing means. A shorter, clearer and stepwise description would be better. From my understanding

a. The mean of the full observation period is taken as ground truth.

b. A model is trained on a short (few months) period

c. The model is used to predict the full observation period.

d. Mean statistics are computed and compared to ground truth. Samples with missing observations are excluded from the calculations.

Thank you for your help in adding clarity and reducing wordiness to the discussion of the MCP process! We have modified Lines 208-217 to read:

"As an initial test of the performance of the algorithms and reference variable combinations, we develop ensembles of MCP-based long-term wind speed estimates at each measurement site using consecutive 12-month training periods, according to the following steps: 1. Establish that 75% of the observations in each month in the training period are available after applying the quality control checks discussed in Section 2.1 (all 66 observations utilised in this work have average and median monthly data recovery and quality rates exceeding the 75% threshold).

2. A model is trained on temporally aligned observation data and reference data during the training period.

3. The model is used to predict the full observation period (Figure 2b).

4. Performance statistics are computed with respect to the observations (Table 1). Timestamps with missing observations are excluded from the statistics."

And Lines 233-236 have been reworded as such:

"The months-long analysis follows the same ensemble formula as the 12-month exercise, just with shorter consecutive training periods. Across the measurement sites, calendar months in the spring and fall had the most single instances of ≥75% data recovery and quality, followed by summer, and lastly winter (Figure 4). Median measurement data recovery and quality percentages according to calendar month ranged from 99.2% (December) to 99.7% (May) (Figure 4)."

3. One of the main reasons why MCP-methods will work well or not on short measurement series is the seasonality of the errors in the reference series. It is not a problem per se if the measurement period is not representative for the long-term as long as the reference series accurately captures the variations. As an extreme example, with perfectly correlated observation and reference series, one observation sample is enough to estimate the true long-term mean wind speed. In practice, as is well demonstrated in the manuscript, the risk of getting poor results are higher if measurements are taken during e.g. low-wind periods. But these phenomena should not be confused (as in e.g. L24-27, L440-442).

Thank you for pointing this out. Per your helpful note, we have reworded to the following:

Lines 24-27: "However, in cases when the shortest observational periods (one to two months) used for correction are not well correlated with the overlapping ERA5 reference, the resultant long-term wind speed errors are worse than those produced using ERA5 without correction."

Lines 484-486: "While even one month of onsite wind speed measurements improves long-term wind speed estimates on average, incorporating at least four months of onsite measurements is a better option to mitigate the errors that could occur if some of the measured and reference wind speeds during the measurement period are poorly correlated."

Lines 491-492: "However, MLR is the least risky algorithm given the possibility of poor correlations between the measurements and the reference data."

4. Energy results should really be in the results section (not in discussion). Since energy production is more important than mean wind speed for the intended application, it would be good to emphasize these results more (on the expense of mean wind speed), e.g. in discussion and abstract.

We appreciate this suggestion to emphasize the energy results, and Reviewer 2 also made this point. We have moved the energy results to a new results subsection: Section 3.4 "Implications for energy production estimates."

Additionally, we have added the following text to the abstract (Lines 32-36): "Translating the analysis to wind energy, median relative errors in the capacity factor are on average within 10% using one month of training. If the observation period used for correction is not well correlated with the reference data, however, misrepresentation of the observed capacity factor can be substantial. The risk associated with poor correlation between the observed and reference datasets decreases with increasing training period length. In the worst correlation scenarios, the median capacity factor relative errors from using one, three, and six months are within 47%, 26%, and 16%, respectively."

**5. It would be good to include descriptions and references of state-of-the-art MCP methods.**

Thank you! We have added the following discussion and references on Lines 175-182:

"One of the advantages of utilising MCP for long-term wind resource estimation is the variety of algorithm choices, which range from simplistic linear regression to machine learning techniques that can be applied to link the short-term and long-term wind speeds. Early MCP methodologies focused on linear (as reported via Rogers et al., 2005: Derrick, 1992; Landberg and Mortenson, 1993; Woods and Watson, 1997; Vermeulen et al., 2001) and quadratic fits (as reported via Rogers et al., 2005: Joensen et al., 1999; Riedel and Strack, 2001). From there, distribution-based probabilistic techniques emerged (García-Rojo, 2004; Sheppard, 2009; Carta and Velázquez, 2011). With the onset of machine learning techniques came applications to MCP-based wind resource analysis, such as using artificial neural networks, support vector machine, and random forest to estimate long-term wind speeds (Díaz et al., 2017)."

Carta, J. A. and Velázquez, S.: A new probabilistic method to estimate the long-term wind speed characteristics at a potential wind energy conversion site, Energy, 36(5) 2671-2685, https://doi.org/10.1016/j.energy.2011.02.008, 2011.

Derrick, A.: Development of the measure-correlate-predict strategy for site assessment, Proceedings of the BWEA, 1992.

Díaz, S. Carta, J. A., and Matías, J. M.: Performance assessment of five MCP models proposed for the estimation of long-term wind turbine power outputs at a target site using three machine learning techniques, Applied Energy, 209, 455-477, https://doi.org/10.1016/j.apenergy.2017.11.007, 2018.

García-Rojo, R.: Algorithm for the estimation of the long-term wind climate at a meteorological mast using a joint probabilistic approach, Wind Engineering, 28, 213-224, https://doi.org/10.1260/0309524041211378, 2004.

Joenson, A., Landberg, L., and Madsen, H.: A new measure-correlate-predict approach for resource assessment, Proceedings of the EWEA, 1999.

Landberg, L. and Mortenson, N. G.: A comparison of physical and statistical methods for estimating the wind resource at a site, Proceedings of the BWEA, 1993.

Riedel, V. and Strack, M.: Robust approximation of functional relationships between meteorological data: alternative measure-correlate-predict algorithms, Proceedings of the EWEA, 2001.

Rogers, A. L., Rogers, J. W., and Manwell, J. F.: Comparison of the performance of four measurecorrelate-predict algorithms, Journal of Wind Engineering and Industrial Aerodynamics, 93(3) 243-264, https://doi.org/10.1016/j.jweia.2004.12.002, 2005.

Sheppard, C. J. R.: Analysis of the measure-correlate-predict methodology for wind resource assessment, Thesis, California State Polytechnic University, https://scholarworks.calstate.edu/concern/theses/9593tx516 , 2009.

Vermeulen, P. E. J., Marijanyan, A., Abrahamyan, A., den Boom J. H.: Application of matrix MCP analysis in mountainous Armenia, Proceedings of the EWEA, 2001.

Woods, J. C. and Watson, S. J.: A new matrix method of predicting long-term wind roses with MCP, J. Wind Eng. Ind. Aerodyn., 66(2), 85-94, https://doi.org/10.1016/S0167-6105(97)00009-3, 1997.

**6. Many different regression tree algorithms exist. State which one you use and give a reference.**

We have modified Lines 194-195 to read: "In this analysis, the ensemble aggregation method used is least-squares boosting with 100 learning cycles per the Matlab algorithm fitrensemble (MathWorks, 2024)."

MathWorks: fitrensemble, https://www.mathworks.com/help/stats/fitrensemble.html, last access: 13 December 2024.

7. The quality control of measurements is performed automatically. This is fine, in particular given the large number of observations. Periods with disturbed observations might however persist after the control, an example is partial icing of the anemometers. This could maybe explain the poorer results in Alaska.

Great point. We have added the following to Lines 355-358: "A potential factor impacting the results for Alaska is the quality of the observations. While the automated quality-control techniques discussed in Section 2.1 remove periods of nonvarying wind speeds due to outages or icing, they may not capture more subtle impacts on the observations, such as partial icing of the anemometers."

**8. L33-34: normally, one describes it as that reference data (e.g. ERA5) are used to correct a short measurement, not the other way around.**

We have reworded the sentence as follows (Lines 38-40): "In the utility-scale wind energy industry, shortterm (less than five years) wind measurements are temporally extended using long-term (decades-long) wind resource simulations to produce a long-term wind energy generation estimate at a site of development interest in an expedient manner."

**9. L58-65 could be rewritten to sound less like a sales text for ArcVera.**

We agree and have reduced the discussion on ArcVera's procedures to the following (Lines 64-66): "For example, ArcVera uses at least one full year of observational data to bias correct their high-resolution model output (ArcVera, 2023)."

**10. Refs to fig 5c should be 5d (L357 + L428)**

Thank you! We have made the changes to 5d and appreciate you catching this typo.

11. L398 "versus performing a bias correction" could be "versus estimating mean wind speed" (the term bias correction should not be reserved for mean wind speed estimates).

We have reworded per your recommendation, thank you.

Lines 400-402: "Additionally, the power production at the low wind sites will be dominated by the tail end of the wind speed distribution, leading to potential significant differences between the skill of the MCP algorithms in reproducing the highest percentiles of wind speeds versus estimating mean wind speeds, as in Sections 3.1-3.3."

**Reviewer 2**

The authors have thoroughly and appropriately addressed the reviewer's comments. I therefore recommend the manuscript for publication and only suggest one smaller technical correction:

• Section 3.4: The heading currently reads "Discussion". I suggest renaming this section for two reasons. First, the title "Discussion" suggests that the discussion of the results is limited to 3.4. However, results are evaluated and discussed in other parts of section 3 as well. Second, new results are presented in subsection 3.4 (e.g. table 3 and figure 12). A new heading should refer to the topic of the sub-section – i.e. "Implications for Capacity Factors/Annual Energy" production or similar.

Thank you for this helpful suggestion and for reviewing the update manuscript! We have renamed Section 3.4 "Implications for energy production estimates." We have also moved the final three paragraphs of the manuscript under a new Section 3.6 "Recommendations and future work." Again, we appreciate your time and your thoughtful review!